



# Is there warming in the pipeline? A multi-model analysis of the zero emission commitment from CO$_2$

Andrew H. MacDougall[1], Thomas L. Frölicher[2,3], Chris D. Jones[4], Joeri Rogelj[5,6], H. Damon Matthews[7], Kirsten Zickfeld[8], Vivek K. Arora[9], Noah J. Barrett[1], Victor Brovkin[10], Friedrich A. Burger[2,3], Micheal Eby[11], Alexey V. Eliseev[12,13], Tomohiro Hajima[14], Philip B. Holden[15], Aurich Jeltsch-Thömmes[2,3], Charles Koven[16], Laurie Menviel[17], Martine Michou[18], Igor I. Mokhov[12,13], Akira Oka[19], Jörg Schwinger[20], Roland Séférian[18], Gary Shaffer[21,22], Andrei Sokolov[23], Kaoru Tachiiri[14], Jerry Tjiputra [20], Andrew Wiltshire[4], and Tilo Ziehn[24]

[1]St. Francis Xavier University, Antigonish, B2G 2W5, Canada
[2]Climate and Environmental Physics, Physics Institute, University of Bern, Switzerland
[3]Oeschger Centre for Climate Change Research, University of Bern, Switzerland
[4]Met Office Hadley Centre, Exeter, EX1 3PB, UK
[5]Grantham Institute for Climate Change and the Environment, Imperial College London, London,UK
[6]International Institute for Applied Systems Analysis (IIASA), Laxenburg, Austria
[7]Concordia University, Montreal, Canada
[8]Department of Geography, Simon Fraser University, Burnaby, Canada
[9]Canadian Centre for Climate Modelling and Analysis, Environment and Climate Change Canada, Victoria, BC, Canada
[10]Max Planck Institute for Meteorology, Hamburg, Germany
[11]University of Victoria, Victoria, BC, Canada
[12]Faculty of Physics, Lomonosov Moscow State University, Moscow, Russia
[13]A.M. Obukhov Institute of Atmospheric Physics, Russian Academy of Sciences, Moscow, Russia
[14]Research Center for Environmental Modeling and Application, Japan Agency for Marine-Earth Science and Technology, Yokohama, Japan
[15]School of Environment, Earth and Ecosystem Sciences, The Open University, Walton Hall, Milton Keynes, MK7 6AA, UK
[16]Climate and Ecosystem Sciences Division, Lawrence Berkeley National Lab, Berkeley, CA, USA
[17]Climate Change Research Centre, PANGEA, The University of New South Wales, Sydney, Australia
[18]CNRM, Université de Toulouse, Météo-France, CNRS, Toulouse, France
[19]Atmosphere and Ocean Research Institute, The University of Tokyo
[20]NORCE Norwegian Research Centre, Bjerknes Centre for Climate Research, Bergen, Norway
[21]Research Center GAIA Antarctica, University of Magallanes, Punta Arenas, Chile
[22]Niels Bohr Institute, University of Copenhagen, Copenhagen, Denmark
[23]Center for Global Change Science, Massachusetts Institute of Technology, Cambridge, USA
[24]Commonwealth Scientific and Industrial Research Organisation, Oceans and Atmosphere, Aspendale, VIC, Australia

**Correspondence:** AH MacDougall (amacdoug@stfx.ca)

**Abstract.** The Zero Emissions Commitment (ZEC) is the change in global mean temperature expected to occur following the cessation of net CO$_2$ emissions, and as such is a critical parameter for calculating the remaining carbon budget. The Zero Emissions Commitment Model Intercomparison Project (ZECMIP) was established to gain a better understanding of the potential magnitude and sign of ZEC, in addition to the processes that underlie this metric. Eighteen Earth system models of both full and intermediate complexity participated in ZECMIP. All models conducted an experiment where atmospheric



CO$_2$ concentration increases exponentially until 1000 PgC has been emitted. Thereafter emissions are set to zero and models are configured to allow free evolution of atmospheric CO$_2$ concentration. Many models conducted additional second priority simulations with different cumulative emissions totals and an alternative idealized emissions pathway with a gradual transition to zero emissions. The inter-model range of ZEC 50 years after emissions cease for the 1000 PgC experiment is -0.36 to 0.29

°C with a model ensemble mean of -0.06°C, median of -0.05°C and standard deviation of 0.19 °C. Models exhibit a wide variety of behaviours after emissions cease, with some models continuing to warm for decades to millennia and others cooling substantially. Analysis shows that both ocean carbon uptake and carbon uptake by the terrestrial biosphere are important for counteracting the warming effect from reduction in ocean heat uptake in the decades after emissions cease. Overall, the most likely value of ZEC on multi-decadal timescales is close to zero, consistent with previous model experiments.

## 1 Introduction

The long-term temperature goal of the Paris Agreement is to hold global warming well below 2°C, and to endeavour to keep warming to no more than 1.5°C (United Nations, 2015). An important metric to assess the feasibility of this target is the 'remaining carbon budget' (e.g. Rogelj et al., 2018), which represents the total quantity of CO$_2$ that can still be emitted without causing a climate warming that exceeds the temperature limits of the Paris Agreement (e.g. Rogelj et al., 2019a). The

remaining carbon budget can be estimated from five factors: 1) historical human induced warming to date; 2) the Transient Climate Response to cumulative CO$_2$ emissions (TCRE); 3) the estimated contribution of non-CO$_2$ climate forcings to future warming; 4) a correction for the feedback processes presently unrepresented by Earth System Models (ESMs); and 5) the unrealized warming from past CO$_2$ emissions, called Zero Emissions Commitment (ZEC) (e.g. Rogelj et al., 2019a). Of these five factors, ZEC is the only quantity whose uncertainty was not formally assessed in the recent Intergovernmental Panel

on Climate Change (IPCC) Special Report on 1.5°C. Here we present the results of a multi-model analysis that uses the output of dedicated model experiments that were submitted to the Zero Emission Commitment Model Intercomparison Project (ZECMIP). This intercomparison project explicitly aims at quantifying the ZEC and identifying the processes that affect its magnitude and sign across models (Jones et al., 2019).

ZEC is the change in global temperature that is projected to occur following a complete cessation of net CO$_2$ emissions

(Matthews and Weaver, 2010). After emissions of CO$_2$ cease, carbon is expected to be redistributed between the atmosphere, ocean, and land carbon pools, such that the atmospheric CO$_2$ concentration continues to evolve over several millennia (e.g. Maier-Reimer and Hasselmann, 1987; Cao et al., 2009; Siegenthaler and Joos, 1992; Sarmiento et al., 1992; Enting et al., 1994; Archer and Brovkin, 2008; Archer et al., 2009; Eby et al., 2009; Joos et al., 2013). In parallel, ocean heat uptake is expected to decline as the ocean comes into thermal equilibrium with the modified radiative forcing (Matthews and Caldeira, 2008). In

previous simulations of ZEC, the carbon cycle has acted to remove carbon from the atmosphere and counteract the warming effect from the reduction in ocean heat uptake, leading to values of ZEC that are close to zero (e.g. Plattner et al., 2008; Matthews and Caldeira, 2008; Solomon et al., 2009; Frölicher and Joos, 2010; Gillett et al., 2011). In the recent assessment of ZEC in the IPCC Special Report on Global Warming of 1.5 °C, the combined available evidence indicated that past CO$_2$





emissions do not commit to substantial further global warming (Allen et al., 2018). A ZEC of zero was therefore applied for the

computation of the remaining carbon budget for the IPCC 1.5 °C Special Report (Rogelj et al., 2018). However, the evidence
available at that time consisted of simulations from only a relatively small number of models using a variety of experimental
designs. Furthermore, some recent simulations have shown a more complex evolution of temperature following cessation of
emissions (e.g. Frölicher et al., 2014; Frölicher and Paynter, 2015). Thus a need to assess ZEC across a wider spectrum of
climate models using a unified experimental protocol has been articulated (Jones et al., 2019).

ZEC was one of the metrics that emerged from the development of ESMs at the turn of the 21$^{st}$ century (Hare and Mein-
shausen, 2006). The concept was first conceptualized by Hare and Meinshausen (2006) who used the Model for the Assessment
of Greenhouse gas Induced Climate Change (MAGICC), a climate model emulator, to explore temperature evolution following
a complete cessation of all anthropogenic emissions. Matthews and Caldeira (2008) introduced the $CO_2$-only concept of ZEC
which is used here. Their experiments used the intermediate complexity University of Victoria Earth System Climate Model

(UVic ESCM) to show that stabilizing global temperature would require near zero $CO_2$ emissions. Plattner et al. (2008) used a
wide range of different Earth System Models of Intermediate Complexity (EMICs) following a similar experiment and found
that ZEC is close to (or less than) zero. These initial results with intermediate complexity models were subsequently supported
by emission-driven ESM simulations (Lowe et al., 2009; Frölicher and Joos, 2010; Gillett et al., 2011). Zickfeld et al. (2013)
quantified the ZEC under different scenarios and for a range of EMICs, but the resulting range is biased towards negative

values as slightly negative instead of zero emissions were prescribed in some models. Some recent ESM simulations indicate
that climate warming may continue after $CO_2$ emissions cease. For example, Frölicher and Paynter (2015) performed a simu-
lation with the full ESM GFDL-ESM2M where emissions cease after 2°C of warming is reached. The simulations show some
decades of cooling followed by a multi-centennial period of renewed warming resulting in an additional 0.5°C of warming
1000 years after emissions cease.

60   Two studies have examined in detail the underlying physical and biogeochemical factors that generate ZEC. Ehlert and
Zickfeld (2017) examine ZEC with a set of idealized experiments conducted with the UVic ESCM. The study partitioned ZEC
into a thermal equilibrium component represented by the ratio of global mean surface air temperature anomaly to unrealized
warming, and a biogeochemical equilibrium component represented by the ratio of airborne fraction of carbon to equilibrium
airborne fraction of carbon. The study found that the thermal equilibrium component of ZEC is much greater than the biogeo-

65   chemical equilibrium component, implying a positive warming commitment. Williams et al. (2017) examine ZEC using the
theoretical framework developed by Goodwin et al. (2007). The framework allows for the calculation of equilibrium atmo-
spheric $CO_2$ concentration if the cumulative effect of the land carbon sink is known. The framework was applied to the same
simulation conducted for Frölicher and Paynter (2015). The analysis showed that ZEC emerges from two competing contri-
butions: 1) a decline in the fraction of heat taken up by the ocean interior leading to radiative forcing driving more surface

70   warming; 2) uptake of carbon by the terrestrial biosphere and ocean system removing carbon from the atmosphere, causing a
cooling effect. Both studies focused on the long-term value of ZEC after multiple centuries and thus neither study examined
what drives ZEC in the policy relevant timeframe of a few decades following cessation of emissions.





While we focus here on the ZEC from $CO_2$ emissions only, the ZEC concept has also been applied to the climate commitment resulting from other greenhouse gas emissions and aerosols (Frölicher and Joos, 2010; Matthews and Zickfeld, 2012; Mauritsen and Pincus, 2017; Allen et al., 2018; Smith et al., 2019), wherein the ZEC is characterized by an initial warming due to the removal of aerosol forcing, followed by a more gradual cooling from the decline in non-$CO_2$ greenhouse gas forcing. The ZEC from all emissions over multiple centuries is generally consistent with ZEC from $CO_2$ emissions only for moderate future scenarios (Matthews and Zickfeld, 2012).

In addition to the ZEC, other definitions of warming commitment have also been used in the literature. The 'constant composition commitment' is defined as the unrealized warming that results from constant atmospheric greenhouse gas and aerosol concentrations (Wigley, 2005; Meehl et al., 2005; Hare and Meinshausen, 2006). This variety of warming commitment was highlighted prominently in the 2007 IPCC report (Meehl et al., 2007), leading to a widespread misunderstanding that this additional 'warming in the pipeline' was the result of past greenhouse gas emissions. However, the constant composition commitment rather results primarily from the future $CO_2$ and other emissions that are required to maintain stable atmospheric concentrations over time (Matthews and Weaver, 2010; Matthews and Solomon, 2013). Another related concept is the future 'emissions commitment' which quantifies the committed future $CO_2$ (and other) emissions that will occur as a result of the continued operation of existing fossil fuel infrastructure (Davis et al., 2010; Davis and Socolow, 2014; Smith et al., 2019; Tong et al., 2019). This concept is also distinct from the ZEC, as it quantifies an aspect of socioeconomic inertia (rather than climate inertia), which has been argued to be an important driver of potentially unavoidable future climate warming (Matthews and Solomon, 2013; Matthews, 2014).

When considering climate targets in the range of 1.5 to 2.0°C and accounting for the approximately 1°C of historical warming to date (Allen et al., 2018; Rogelj et al., 2018), a ZEC on the order of ± 0.1°C can make a large difference in the remaining carbon budget. Hence there is a need for a precise quantification and in-depth understanding of this value. This can be achieved by a systematic assessment of ZEC across the range of available models and a dedicated analysis of the factors that control the value of ZEC in these simulations. Thus the goals of this study based on the simulations of the Zero Emissions Commitment Model Intercomparison Project (ZECMIP) are: 1) to estimate the value of ZEC in the decades following cessation of emissions in order to facilitate an estimate of the remaining carbon budget; 2) to test if ZEC is sensitive to the pathway of emissions; 3) to establish whether ZEC is dependent on the cumulative total $CO_2$ that are emitted before emissions cease; 4) to identify which physical and biogeochemical factors control the sign and magnitude of ZEC in models.

The most policy relevant question related to ZEC is: will global temperature continue to increase following complete cessation of greenhouse gas and aerosol emissions? The present iteration of ZECMIP aims to answer part of this question by examining the temperature response in idealized $CO_2$-only climate model experiments. To answer the question in full, the behaviour of non-$CO_2$ greenhouse gases, aerosols, and land-use-change must be accounted for in a consistent way. Such efforts will be the focus of future iterations of ZECMIP.



## 2 Methods

### 2.1 Protocol and Simulations

Here we summarize the ZECMIP protocol, the full protocol for ZECMIP is described in Jones et al. (2019). The ZECMIP protocol requested modelling groups to conduct three idealized simulations of two different types each - A and B. Type A simulations are initialized from one of the standard climate model benchmark experiments in which specified atmospheric $CO_2$ concentration increases at a rate of 1% per year from its pre-industrial value of around 285 ppm until quadrupling, referred to as the 1pctCO2 simulation in the Climate Model Intercomparison Project (CMIP) framework Eyring et al. (2016). The three type A simulations are initialized from the 1pctCO2 simulation when diagnosed cumulative emissions of $CO_2$ reach 750, 1000, and 2000 Pg C. After the desired cumulative emission are reached the models are set to freely evolving atmospheric $CO_2$ mode, with zero further $CO_2$ emissions. Since net anthropogenic emissions are specified to be zero in type A simulations, atmospheric $CO_2$ concentration is expected to decline in these simulations in response to carbon uptake by oceans and land. A consequence of the protocol is that for the type A simulations each model branches from the 1pctCO2 simulations in a different year, contingent on when a model reaches the target cumulative emissions, which in turn depends on each model's representation of the carbon cycle and feedbacks. An example of emissions for the type A experiments is shown in Figure 1a. The three type B simulations are initialized from pre-industrial conditions and are emissions driven from the beginning of the simulation. Emissions follow bell-shaped pathways wherein all emissions occur within a 100 year window (Figure 1b). In all experiments land-use-change and non-$CO_2$ forcings are held at their pre-industrial levels.

Due to the late addition of ZECMIP to the CMIP Phase 6 (CMIP6) (Eyring et al., 2016), only the 1000 Pg C type A experiment (esm-1pct-brch-1000PgC) was designated as top priority ZECMIP simulation. The other simulations were designated as second priority simulations and were meant to be conducted if participating modelling groups had the resources and time. Both full ESMs and EMICs were invited to participate in ZECMIP. ESMs were requested to perform the top priority simulation for 100 years after $CO_2$ emissions cease, and more years and more experiments as resources allowed. EMICs were requested to conduct all experiments for at least 1000 years of simulations following cessation of emissions. Table 1 shows the experiments and experimental codes for ZECMIP.

**Table 1.** Experiments designed for ZECMIP

| Name | Code | Cumulative Emissions (PgC) | Priority |
|------|------|---------------------------|----------|
| A1 | esm-1pct-brch-1000PgC | 1000 | 1 |
| A2 | esm-1pct-brch-750PgC | 750 | 2 |
| A3 | esm-1pct-brch-2000PgC | 2000 | 2 |
| B1 | esm-bell-1000PgC | 1000 | 2 |
| B2 | esm-bell-750PgC | 750 | 2 |
| B3 | esm-bell-2000PgC | 2000 | 2 |





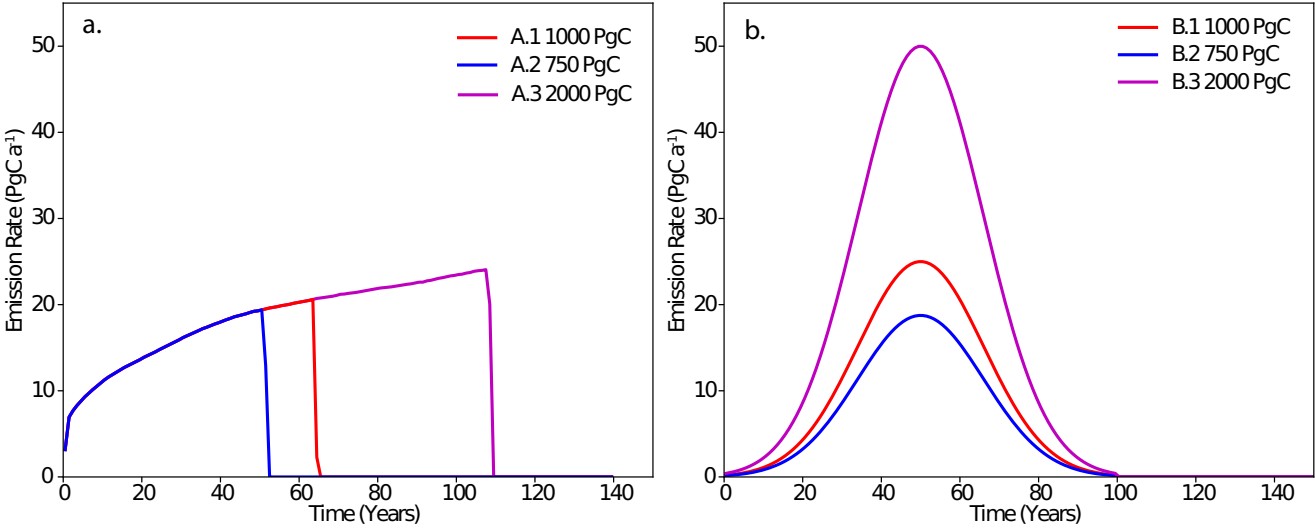

**Figure 1.** (a) Example of diagnosed emission from the UVic ESCM for the type A experiments. Emissions are diagnosed from the 1pctCO2 experiment which has prescribed atmospheric $CO_2$ concentrations. The target cumulative emissions total is reach part-way through the final year of emissions, thus that final year has a lower average emission rate than the previous year. (b) Time series of global $CO_2$ emissions for bell curve pathways B1 to B3. The numbers in the legend indicate the cumulative amount of $CO_2$ emissions for each simulation.

## 2.2 Model Descriptions

Eighteen models participated in ZECMIP: nine comprehensive ESMs and nine EMICs. The primary features of each model are summarized in Table 2 and 3 for ESMs, and Table 4 and 5 for EMICs. The ESMs in alphabetical order are: 1) CSIRO Australian Community Climate and Earth System Simulator, ESM version 1.5 – ACCESS-ESM1.5, 2) Canadian Centre for Climate Modelling and Analysis (CCCma) – CanESM5, 3) Community Earth System Model 2 – CESM2, 4) Centre National de Recherches Météorologiques (CNRM) – CNRM-ESM2-1, 5) Geophysical Fluid Dynamics Laboratory (GFDL) – GFDL-

ESM2M, 6) Japan Agency for Marine-Earth Science and Technology (JAMSTEC/team MIROC) – MIROC-ES2L, 7) Max Planck Institute Earth System model, version 1.2, low resolution – MPI-ESM1.2-LR, 8) Norwegian Earth System Model 2 – NorESM2 and 9) UK (Met Office Hadley Centre and NERC) – UKESM1-0-LL. The nine EMICs in alphabetical order are: 1) Bern three dimensional Earth System Model – Bern3D-LPX, 2) Climate-Biosphere model, version 2 - CLIMBER-2, 3) Danish Centre for Earth System Science Earth System Model – DCESS1.0, 4) A.M. Obukhov Institute of Atmospheric Physics,

Russian Academy of Sciences – IAPRAS, 5) Loch-Vecode-ECbilt-Clio Model – LOVECLIM 1.2, 6) Massachusetts Institute of Technology Earth System Model – MESM, 7) Model for Interdisciplinary Research on Climate-lite/Japan Uncertainty Modelling Project-Loosely Coupled Model – MIROC-lite, 8) Planet Simulator - Grid-Enabled Integrated Earth system model – PLASIM-GENIE, and 9) University of Victoria Earth System Climate Model – UVic ESCM 2.9pf. For brevity models are referred to by their short names in the remainder of the manuscript. Table 6 shows the ZECMIP experiments that each

modelling group submitted.





Table 7 shows three benchmark climate metrics for each model, Equilibrium Climate Sensitivity (ECS), Transient Climate Response (TCR), and TCRE. ECS is the climate warming expected if atmospheric $CO_2$ concentration was doubled from the pre-industrial value and maintained indefinitely while the climate system is allowed to come into equilibrium with the elevated radiative forcing (e.g. Planton, 2013). There are a variety of methods to compute ECS from climate model outputs (e.g. Knutti

et al., 2017). Here we use ECS values computed using the method of Gregory et al. (2004), often called Gregory ECS or "effective climate sensitivity". The method of Gregory et al. (2004) computes ECS from the slope of the scatter plot between change in global temperature and planetary heat uptake, with values from the benchmark experiment where atmospheric $CO_2$ concentration is instantaneously quadrupled (4×CO2 experiment). Usually the slope is computed from year 20 to year 140 of the 4×CO2 experiment (Andrews et al., 2012). TCR is the climate warming (relative to the preindustrial temperature) when

atmospheric $CO_2$ is doubled in year 70 of the 1pctCO2 experiment, computed using a 20 year averaging window centred about year 70 of the experiment (e.g. Planton, 2013). TCRE is described in the introduction and is computed from year 70 of the 1% experiment (e.g. Planton, 2013).

Bern and UVic submitted three versions of their models with three different ECSs. For Bern ECSs of 2.0 °C, 3.0 °C, and 5.0 °C, and for UVic ECSs of 2.0 °C, 3.5 °C, and 4.5 °C. These ECS values are true equilibrium climate sensitivity computed by

allowing each model to come fully into equilibrium with the changed radiative forcing. For each model the central ECS value was used for the main analysis, 3.0 °C for Bern and 3.5 °C for UVic. The remaining experiments were used to explore the relationship between ECS and ZEC.



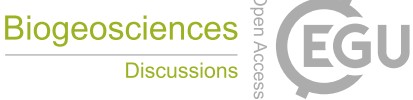

**Table 2.** Model descriptions of the atmospheric, oceanic, and carbon cycle components for the full Earth System Models (ESMs) that participated in this study.

| Model | ACCESS-ESM1.5 | CanESM5 | CESM2 | CNRM-ESM2-1 | GFDL ESM2M |
|---|---|---|---|---|---|
| Short Name | ACCESS | CanESM5 | CESM2 | CNRM | GFDL |
| Model Expansion | CSIRO Australian Community Climate and Earth System Simulator, ESM version 1.5 | Canadian Earth System Model, version 5 | Community Earth System Model 2 | CNRM-CERFACS Earth System Model, version 2 | Geophysical Fluid Dynamics Laboratory Earth system model version 2 |
| Atmosphere | Unified Model (UM) 7.3, 1.875°x1.25°, L38 | CanAM5, 2.81° x 2.81°, L49 | CAM6, 0.9°x1.25° | ARPEGE-Climat T127 ( 1.4°), 91 levels | AM2, 2° x 2.5°, L24 |
| Ocean | MOM5, 1° tripolar grid, finer 10S-10N and S. Ocean, L50 | NEMO, 1° finer 20°N - 20°S, L45 | POP2 | NEMO, 1° tripolar grid, L75 | MOM4p1, 1° tripolar grid finer at equator, L50 |
| Sea Ice | CICE4.1 | LIM2 | POP2 | GELATOv6 | SIS |
| Z-coordinate or Isopycnal | Z-coordinate | Z-coordinate | Z-coordinate | Z-coordinate | Z-coordinate |
| Land Carbon Cycle | | | | | |
| Model Name | CABLE | CLASS-CTEM | CLM5 | SURFEX (ISBA-CTRIP) | LM3.0 |
| Dynamic Vegetation | No | No | No | No | Yes |
| Nitrogen Cycle | Yes | No | Yes | No | No |
| Phosphorus Cycle | Yes | No | No | No | No |
| Permafrost Carbon | No | No | Yes | No | No |
| Ocean Carbon Cycle | | | | | |
| Model Name | WOMBAT | CMOC | MARBL | PISCESv2-gas | TOPAZ2 |
| Explicit Nutrients | Yes | Yes | yes | yes | Yes |
| If Yes List | P, Fe | N | N, P, Si, Fe | N, P, Si, Fe | N, P, Si, Fe |
| Reference | (Law et al., 2017; T. et al.) | (Swart et al., 2019) | (Danabasoglu and Others, 2019; Lawrence et al., 2019) | (Séférian et al., 2019; Decharme et al., 2019; Delire et al., 2020) | (Dunne et al., 2012, 2013) |



**Table 3.** Model descriptions of the atmospheric, oceanic, and carbon cycle components for the full Earth System Models (ESMs) that participated in this study.

| Model | MIROC-ES2L | MPI-ESM1.2-LR | NorESM2-LM | UKESM1-0-LL |
|---|---|---|---|---|
| Short Name | MIROC-ES2L | MPI-ESM | NorESM2 | UKESM |
| Model Expansion | Model for Interdisciplinary Research on Climate, Earth System version2 for Long-term simulations | Max-Planck-Insitute Earth System model, version 1.2, low resolution | Norwegian Earth System Model 2 | United Kingdom Earth System Model, vn1 |
| Atmosphere | CCSR-NIES AGCM, T42, L40 | ECHAM6, T63 (ca.1.8°x1.8°), L47 | CAM6, 0.9°x2.5°, L32 | HadGAM3. N96 (1.25° x 1.875°), L85 |
| Ocean | CCSR Ocean Component model (COCO), 360x256 grids with tripolar grid, L62 | MPIOM1.6, GR1.5 (1.5°x1.5°) | Bergen Layered Ocean Model, 1° finer near Equator, L53 | NEMO, 1° tripolar grid, L75 |
| Sea Ice | COCO | Thermodynamic – Dynamic | Community Sea-ice model | CICE sea ice model |
| Z-coordinate or Isopycnal | Z-coordinate | Z-coordinate | Isopycnal | Z-coordinate |
| Land Carbon Cycle | | | | |
| Model Name | MATSIRO/ VISIT-e | JSBACH 3.2 | CLM5 | JULES |
| Dynamic Vegetation | No | Yes | No | Yes |
| Nitrogen Cycle | Yes | Yes | Yes | Yes |
| Phosphorus Cycle | No | No | No | No |
| Permafrost Carbon | No | No | Yes | No |
| Ocean Carbon Cycle | | | | |
| Model Name | OECO2 | HAMOCC6 | iHAMOCC | MEDUSA-2 |
| Explicit Nutrients | Yes | Yes | Yes | Yes |
| If Yes List | N, P, Fe | N, P, Si, Fe | P, N, Fe | N, Si, Fe |
| Reference | (Hajima et al., 2019) | (Ilyina et al., 2013; Mauritsen et al., 2019; Goll et al., 2017) | (Tjiputra et al., 2020) | (Sellar et al., 2019; Best et al., 2011; Clark et al., 2011; Yool et al., 2013) |



**Table 4.** Model descriptions of the atmospheric, oceanic, and carbon cycle components for Earth system models of intermediate complexity (EMICs) that participated in this study.

| Model | Bern3D-LPX | CLIMBER-2 | DCESS | IAPRAS | LOVECLIM 1.2 |
|---|---|---|---|---|---|
| Short Name | Bern | CLIMBER | DCESS | IAPRAS | LOVECLIM |
| Model Expansion | Bern3D-LPX | Climate-Biosphere model, version 2 | Danish Center for Earth System Science Earth System Model version 1.0 | A.M. Obukhov Institute of Atmopsheric Physics, Russian Academy of Sciences | LOVECLIM v1.2 |
| Atmosphere | 2D Energy-Moisture Balance, $4.5° \ 9°$ (on average) | Statistical-Dynamical, $51° \times 10°$ | Energy-Moisture Balance model | Statistical-Dynamical model, $4.5° \ 6.0°$, L11 | ECBilt, $5.625° \times 5.625°$, L3 |
| Ocean | $4.5° \times 9°$ (on average), L32 | 2D, 3-basin zonally-averaged, $2.5°$ lat, L21 | 2 box in lat, 100 m Z resolution | Statistical-Dynamical model $4.5° \ 6.0°$, L3 | CLIO, $3° \ 3°$, L20 |
| Sea Ice | $4.5°, 9°$ (on average) | Thermodynamic-Dynamic | Mean Surface Temperature Parameterization | Mean Surface Temperature Parameterization | Thermodynamic-dynamic |
| Z-coordinate or isopycnal | Z-coordinate | Z-coordinate | Z-coordinate | Z-coordinate | Z-coordinate |
| Land Carbon Cycle | | | | | |
| Model Name | LPX v1.4 | VECODE | | | VECODE |
| Dynamic Vegetation | Yes | Yes | Yes | No | Yes |
| Nitrogen Cycle | Yes | No | No | No | No |
| Phosphorus Cycle | No | No | No | No | No |
| Permafrost Carbon | No | No | No | No | No |
| Ocean Carbon Cycle | | | | | |
| Model Name | Bern3D v2.0s | | | | LOCH |
| Explicit Nutrients | Yes | Yes | Yes | No | Yes |
| If Yes List | P, Si, Fe | P | P | | P, Si |
| Reference | (Ritz et al., 2011; Roth et al., 2014; Jeltsch-Thömmes et al., 2019; Lienert and Joos, 2018) | (Brovkin et al., 2002; Ganopolski et al., 2001) | (Shaffer et al., 2008) | (Eliseev, 2011; Eliseev and Mokhov, 2011; Mokhov and Eliseev, 2012; Eliseev et al., 2014; Mokhov et al., 2020) | (Menviel et al., 2008; Goosse et al., 2010; Mouchet, 2011) |





**Table 5.** Model descriptions of the atmospheric, oceanic, and carbon cycle components for Earth system models of intermediate complexity (EMICs) that participated in this study.

| Model | MESM | MIROC-lite/JUMP-LCM | PLASIM-GENIE | UVic ESCM 2.9pf |
|---|---|---|---|---|
| Short Name | MESM | MIROC-lite | P. GENIE | UVic |
| Model Expansion | MIT Earth System Model | Model for Interdisciplinary Research on Climate-lite/Japan Uncertainty Modelling Project-Loosely Coupled Model | Planet Simulator - Grid-ENabled Integrated Earth system model | University of Victoria Earth System Climate Model version 2.9 permafrost |
| Atmosphere | Zonally averaged 4° lat., L11 | 2D Energy-Moisture Balance, 6°x 6° | PLASIM, T21, L10 | 2D Energy-Moisture Balance, 3.6°1.8°,L1 |
| Ocean | 4°x5° mixed layer, anomaly diffusing model | CCSR Ocean Component model (COCO), 6°x 6°, L15 | GOLDSTEIN, T21, L16 | MOM2, 3.6°1.8°,L19 |
| Sea Ice | Thermodynamic-Dynamic | COCO | GOLDSTEIN | Thermodynamic-Dynamic |
| Z-coordinate or isopycnal | NA | Z-coordinate | Z-coordinate | Z-coordinate |
| **Land Carbon Cycle** | | | | |
| Model Name | TEM | Sim-CYCLE | ENTS | TRIFFID |
| Dynamic Vegetation | No | No | Yes | Yes |
| Nitrogen Cycle | Yes | No | No | No |
| Phosphorus Cycle | No | No | No | No |
| Permafrost Carbon | No | No | No | Yes |
| **Ocean Carbon Cycle** | | | | |
| Model Name | OCM | | BIOGEM | 2NZPD |
| Explicit Nutrients | No | Yes | Yes | Yes |
| If Yes List | | N | P | N, P |
| Reference | (Sokolov et al., 2018) | (Tachiiri et al., 2010; Oka et al., 2011) | (Holden et al., 2018, 2019) | (Weaver et al., 2001; Eby et al., 2009; MacDougall and Knutti, 2016) |





**Table 6.** Experiments conducted for ZECMIP by model. Full ESMs are listed on top followed by EMICs.

| Model | A1 | A2 | A3 | B1 | B2 | B3 |
|---|---|---|---|---|---|---|
| ACCESS | X | X | X | – | – | – |
| CanESM5 | X | – | X | – | – | – |
| CESM2 | X | – | – | – | – | – |
| CNRM | X | – | – | – | – | – |
| GFDL | X | X | X | X | X | X |
| MIROC-ES2L | X | X | X | – | – | – |
| MPI-ESM | X | – | – | – | – | – |
| NorESM2 | X | – | – | – | – | – |
| UKESM | X | X | X | – | – | – |
| Bern | X | X | X | X | X | X |
| CLIMBER | X | – | – | – | – | – |
| DCESS | X | X | X | X | X | X |
| IAPRAS | X | X | X | X | X | X |
| LOVECLIM | X | X | – | X | – | – |
| MESM | X | X | X | X | X | X |
| MIROC-lite | X | X | X | X | X | X |
| P. GENIE | X | X | X | X | X | X |
| UVic | X | X | X | X | X | X |

**Table 7.** Benchmark climate model characteristics for each model: Equilibrium Climate Sensitivity (ECS), Transient Climate Response (TCR), and Transient Climate Response to Cumulative $CO_2$ Emissions (TCRE). UKESM reported a maximum to minimum range for TCR and TCRE based on four ensemble members.

| Model | ECS (°C) | TCR (°C) | TCRE (K EgC$^{-1}$) |
|---|---|---|---|
| ACCESS | 3.9 | 1.9 | 1.9 |
| CanESM5 | 5.7 | 2.8 | 1.9 |
| CESM2 | 5.1 | 2.0 | 1.9 |
| CNRM | 4.8 | 1.9 | 1.7 |
| GFDL | 2.4 | 1.4 | 1.03 |
| MIROC-ES2L | 2.7 | 1.5 | 1.3 |
| MPI-ESM | 2.8 | 1.8 | 1.6 |
| NorESM2 | 2.6 | 1.5 | 1.4 |
| UKESM | 5.4 | 2.68 to 2.85 | 2.49 to 2.66 |
| Bern | 2.6 | 1.6 | 1.6 |
| CLIMBER | 2.8 | 1.8 | 1.7 |
| DCESS | 3.0 | 2.0 | 2.0 |
| IAPRAS | 2.2 | 1.5 | 1.5 |
| LOVECLIM | 2.8 | 1.5 | 1.3 |
| MESM | 2.9 | 1.8 | 1.7 |
| MIROC-lite | 1.7 | 1.2 | 1.03 |
| P. GENIE | 3.4 | 1.7 | 1.4 |
| UVic | 3.5 | 1.8 | 1.7 |



## 2.3 Quantifying ZEC

ZEC is the change in global average surface air temperature following the cessation of $CO_2$ emissions. Thus ZEC must be
calculated relative to the global temperature when emissions cease. Typically such a value would be computed from a 20 year window centred on the year when emissions cease. However, for the ZECMIP type A experiments such a calculation underestimates the temperature of cessation, due to the abrupt change in forcing when emissions suddenly cease, leading to an overestimation of ZEC values. That is, a roughly linear increase in temperature pathway abruptly changes to a close to stable temperature pathway. Therefore we define the temperature of cessation to be the global mean surface air temperature
from benchmark 1pctCO2 experiment averaged over a 20 year window centred on the year emissions cease in the respective ZECMIP type-A experiment (year the ZECMIP experiment branches from the 1% experiment). This method provides an unbiased estimate of the temperature of cessation in the EMICs which lack internal variability, that is the uncertainty range of the temperature of cessation is relatively low.

Earlier studies have examined ZEC from decadal (Matthews and Zickfeld, 2012; Mauritsen and Pincus, 2017; Allen et al.,
2018; Smith et al., 2019) to multi-centennial timescales (Frölicher and Paynter, 2015; Ehlert and Zickfeld, 2017). One of the main motivations of this present study is to inform the impact of the ZEC on the remaining carbon budget. This remaining carbon budget is typically used to assess the consistency of societal emissions pathways with the international temperature target of the Paris Agreement (UNEP, 2018). In this emission pathway and policy context, the ZEC within a few decades of emissions cessation is more pertinent than the evolution of the Earth system hundreds or thousands of years into the future.
Therefore, we define values of $ZEC_X$ as a 20 year average temperature anomaly centred at year $X$ after emissions cease. Thus, 50-year ZEC ($ZEC_{50}$) is the global mean temperature relative to the temperature of cessation averaged from year 40 to year 59 after emissions cease. We similarly define 25-year ZEC ($ZEC_{25}$) and 90-year ZEC ($ZEC_{90}$).

## 2.4 Analysis Framework

A key question of the present study is why some models have positive ZEC and some models have negative or close to
zero ZEC. From elementary theory we understand that the sign of ZEC will depend on the pathway of atmospheric $CO_2$ concentration and ocean heat uptake following cessation of emissions (Wigley and Schlesinger, 1985). Complicating this dynamic is that atmospheric $CO_2$ change has contributions both from the net carbon flux from the ocean and the terrestrial biosphere. Using the forcing-response equation (Wigley and Schlesinger, 1985) and the common logarithmic approximation for the radiative forcing from $CO_2$ (Myhre et al., 1998), we can partition ZEC into contributions from ocean heat uptake, ocean
carbon uptake, and net carbon flux into the terrestrial biosphere. The full derivation of the relationship is shown in Appendix A and the summary equations are shown below:

$$\lambda T_{ZEC} = -R \int_{t=ze}^{\infty} \frac{f_O}{C_A} dt - R \int_{t=ze}^{\infty} \frac{f_L}{C_A} dt - \epsilon(N - N_{ze}), \tag{1}$$





where $\lambda$ (W m$^{-2}$K$^{-1}$) is the climate feedback parameter, $T_{ZEC}$ (K) is ZEC, $R$ (W m$^{-2}$) is the radiative forcing from an e-fold increase in atmospheric CO$_2$ burden, $t$ is time (a), $ze$ is the time emissions cease, $f_O$ (PgC a$^{-1}$) is ocean carbon uptake,

$f_L$ (PgC a$^{-1}$) is carbon uptake by land, $C_A$ is atmospheric CO$_2$ content (PgC), $N$ is planetary heat uptake (W m$^{-2}$), $N_{ze}$ is planetary heat uptake at the time emissions cease, and $\epsilon$ is the efficacy of planetary heat uptake. The equation states that ZEC is proportional to the sum of three energy balance terms: 1) the change in radiative forcing from carbon taken up by the ocean; 2) the change in radiative forcing from carbon taken up or given off by land; and 3) the change in ocean heat uptake. The two integral terms can be evaluated numerically from the ZECMIP model output, and thus can be simplified into two energy

forcing terms $F_{ocean}$ and $F_{land}$:

$$F_{ocean} = R \int\limits_{t=ze}^{\infty} \frac{f_O}{C_A} dt, \tag{2}$$

and,

$$F_{land} = R \int\limits_{t=ze}^{\infty} \frac{f_L}{C_A} dt, \tag{3}$$

and thus:

$$\lambda T_{ZEC} = -F_{ocean} - F_{land} - \epsilon(N - N_{ze}). \tag{4}$$

Values for $R$ were computed from the effective radiative forcing value for the models that simulate internal variability, with effective radiative forcing provided by each modelling group. Bern, DCESS, and UVic prescribe exact values for $R$ and thus these values were used for calculations with these models. Effective radiative forcing is $\frac{1}{2}$ the y-intercept of a 4×CO2 Gregory plot (Gregory et al., 2004). $R$ values and the Gregory ECSs were used to calculate $\lambda$ for each model. Efficacy (Winton et al.,

2010) was calculated from:

$$\epsilon = \frac{\lambda T - R\ln\left(\frac{C_A}{C_{Ao}}\right)}{N}, \tag{5}$$

where C$_{Ao}$ (PgC) is the pre-industrial CO$_2$ burden, and $T$ (K) is the global mean temperature anomaly relative to pre-industrial temperature. $T$, $N$, and $C_A$ values were taken from the benchmark 1pctCO2 for each model, as an average value from year 10 to year 140 of that experiment. Computed $\epsilon$ values are shown in Table 8. In CLIMBER planetary or ocean heat uptake is not

included into standard output and hence is not analyzed using this framework. CESM2 and NorESM2 are also excluded as the 4×CO2 experiment results for these models are not yet available.

Notably, calculated Gregory ECSs and effective radiative forcing vary slightly within models due to the internal variability of the models (Gregory et al., 2015), hence the efficacy values calculated here are associated with some uncertainty. Efficacy





values are known to evolve in time (Winton et al., 2010), thus the efficacy value from the 1pctCO2 experiment may be different

than efficacy 50 years after emission cease in the ZECMIP experiments. To test this effect yearly efficacy values were calculated

for the four EMICs without internal variability (Bern, DCESS, MESM and UVic). These tests showed that efficacy was 3.5%

to 25% away from the values for the 1pctCO2 experiment 50 years after emission cease (Figure A1). Thus we have assigned

efficacy a $\pm$ 30% uncertainty. Radiative forcing from $CO_2$ is not precisely logarithmic (Gregory et al., 2015; Byrne and

Goldblatt, 2014; Etminan et al., 2016) and therefore the calculated $F_{ocean}$ and $F_{land}$ values will be slightly different than the

changes in radiative forcing experienced within each model, except for the three models that prescribe $CO_2$ radiative forcing.

The difference in $CO_2$ radiative forcing between the Myhre et al. (1998) and Etminan et al. (2016) parameterization are about

5 to 7% for $CO_2$ concentration range simulated in the A1 ZECMIP experiment. As Etminan et al. (2016) used new line-by-line

absorption data to compute radiative forcing, existing ESMs which internally compute $CO_2$ likely lie between the Myhre et al.

(1998) and Etminan et al. (2016) parameterizations (Etminan et al., 2016). Also accounting for the uncertainty in recovering $R$

values from model output, we assign a $\pm 10\%$ uncertainty to radiative forcing values.

**Table 8.** Efficacy $\epsilon$ and radiative forcing for $2 \times CO_2$ $R$ values for each model. Efficacy values are calculated from the 1pctCO2 experiment.

| Model | Efficacy | Radiative forcing $2 \times CO_2$ (W m$^{-2}$) |
|---|---|---|
| ACCESS | 1.2 | 2.9 |
| CanESM5 | 1.0 | 3.4 |
| CNRM | 0.9 | 3.2 |
| GFDL | 1.3 | 3.6 |
| MIROC-ES2L | 1.0 | 4.1 |
| MPI-ESM | 1.1 | 4.1 |
| UKESM | 1.0 | 4.0 |
| Bern | 1.0 | 3.7 |
| DCESS | 1.1 | 3.7 |
| IAPRAS | 1.1 | 3.7 |
| LOVECLIM | 1.0 | 3.7 |
| MESM | 0.8 | 4.1 |
| MIROC-lite | 1.0 | 3.0 |
| P. GENIE | 0.9 | 4.2 |
| UVic | 1.0 | 3.7 |

## 3 Results

### 3.1 A1 Experiment results

Figure 2 shows the evolution of atmospheric $CO_2$ concentration and temperature for the 100 years after emissions cease for

the A1 experiment (1% branched at 1000 PgC). In all simulations atmospheric $CO_2$ concentration declines after emissions



cease, with a rapid decline in the first few decades followed by a slower decline thereafter. The rates of decline vary across the models. By 50 years after emissions cease in the A1 experiment the change in atmospheric $CO_2$ concentration ranged from -91 to -51 ppm, with a mean of -76 ppm and median of -80 ppm. Temperature evolution in the 100 years following cessation of emissions varies strongly by model, with some models showing declining temperature, others having ZEC close to zero, and others showing continued warming following cessation of emissions. Some models such as UKESM, CNRM, and

UVic exhibit continued warming in the century following cessation of emissions. Other models such as IAPRAS and DCESS exhibit a temperature peak, then decline. Still other models show ZECs that hold close to zero (e.g. MPI-ESM), while some models show continuous decline in temperature following cessation of emissions (e.g. P. GENIE). Table 9 shows the $ZEC_{25}$, $ZEC_{50}$, and $ZEC_{90}$ values for the A1 experiment. The table shows values of $ZEC_{50}$ ranging from -0.36 to 0.29 °C with a model ensemble mean of -0.06°C, median of -0.05°C and a standard deviation of 0.19°C. Tables A1 and B1 show $ZEC_{25}$, $ZEC_{50}$,

and $ZEC_{90}$ for the A2 and A3 experiment. Figure 3 shows the evolution of atmospheric $CO_2$ concentration, temperature anomalies (relative to the year emissions cease), and ocean heat uptake for 1000 years following cessation of emissions in the A1 experiment. All models show continued decline in atmospheric $CO_2$ concentration for centuries after emissions cease. One model (P. GENIE) shows a renewed growth in atmospheric $CO_2$ concentration beginning about 600 years after emissions cease. The renewed growth in atmospheric $CO_2$ in P. GENIE results from release of carbon from soils overwhelming the residual

ocean carbon sink. Of the eight models that extended simulations beyond 150 years, five show temperature peaking then declining (Bern, MESM, DCESS, LOVECLIM and MIROC-ES2L), GFDL shows temperature declining and then increasing but ultimately remaining close to the temperature at cessation, and the UVic model shows continuous, if slow, warming. Most models show continuous decline in ocean heat uptake with values approaching zero. Three models (GFDL, LOVECLIM and IAPRAS) show the ocean transition from a heat sink to a heat source.



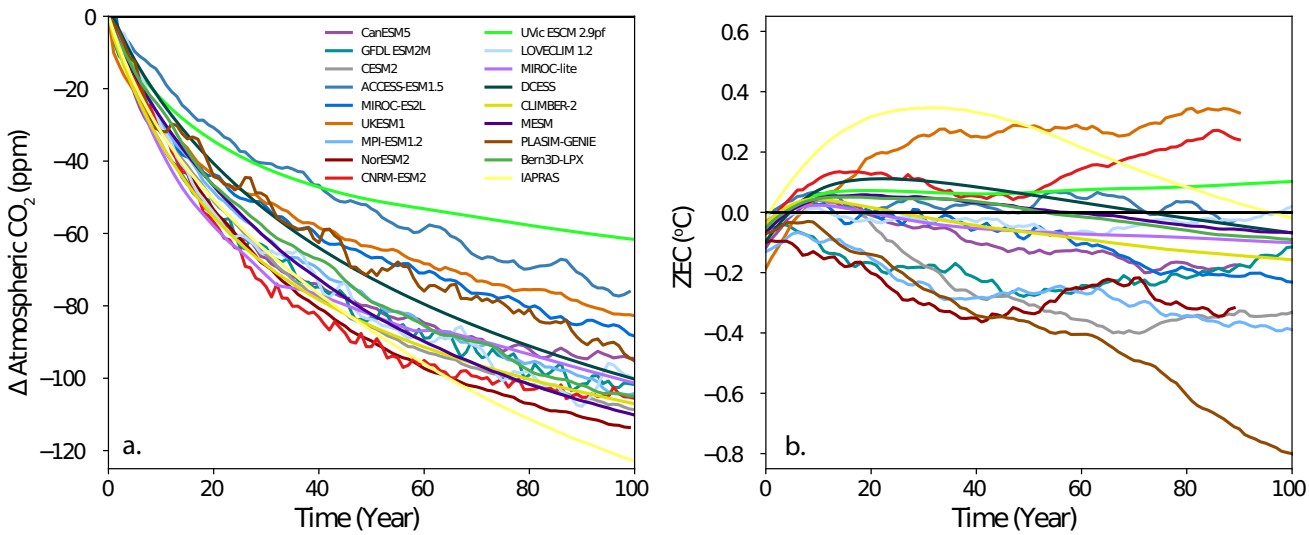

**Figure 2.** (a) Atmospheric $CO_2$ concentration anomaly and (b) Zero Emissions Commitment following cessation of emissions under the experiment where 1000 PgC was emitted following the 1% experiment (A1). ZEC is the temperature anomaly relative to the estimated temperature at the year of cessation. ZEC has been smoothed with a 20 year moving window.

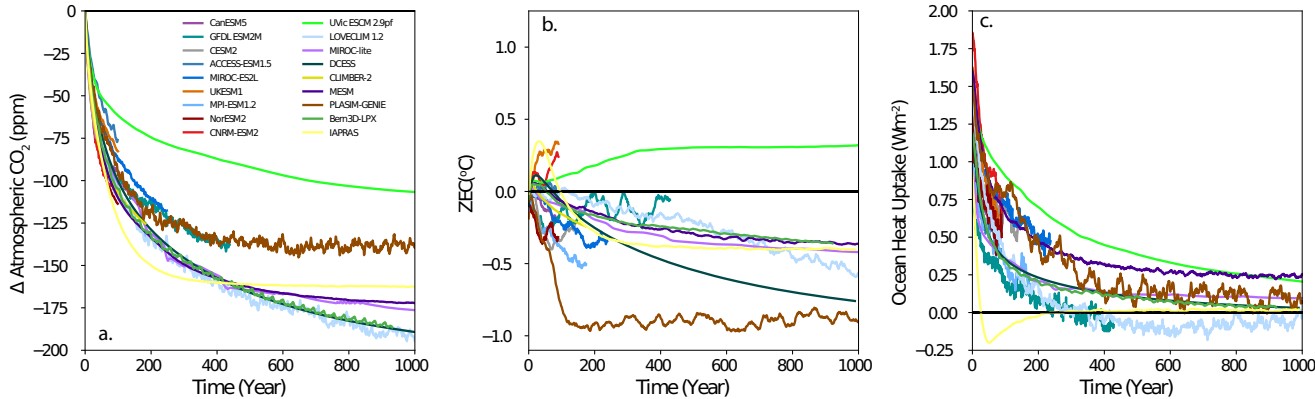

**Figure 3.** (a) Change in atmospheric $CO_2$ concentration, (b) change in temperature, and (c) ocean heat uptake following cessation of emissions for the A1 experiment (1000 PgC following 1%) for 1000 years following cessation of emissions.





**Table 9.** Temperature anomaly relative to the year emissions cease averaged over a 20 year time window centred about the 25th, 50th, and 90th year following cessation of anthropogenic $CO_2$ emissions ($ZEC_{25}$, $ZEC_{50}$, and $ZEC_{90}$ respectively) for the A1 (1000 PgC 1% experiment).

| Model | $ZEC_{25}$ (°C) | $ZEC_{50}$ (°C) | $ZEC_{90}$ (°C) |
|---|---|---|---|
| ACCESS | 0.04 | 0.01 | -0.03 |
| CanESM5 | -0.04 | -0.13 | -0.17 |
| CESM2 | -0.11 | -0.31 | -0.34 |
| CNRM | 0.11 | 0.06 | 0.25 |
| GFDL | -0.18 | -0.27 | -0.19 |
| MIROC-ES2L | -0.02 | -0.08 | -0.21 |
| MPI-ESM | -0.22 | -0.27 | -0.37 |
| NorESM | -0.27 | -0.33 | -0.32 |
| UKESM | 0.21 | 0.28 | 0.33 |
| Bern | 0.05 | 0.01 | -0.08 |
| DCESS | 0.11 | 0.06 | -0.04 |
| CLIMBER | 0.0 | -0.07 | -0.14 |
| IAPRAS | 0.34 | 0.29 | 0.03 |
| LOVECLIM | -0.02 | -0.04 | -0.03 |
| MESM | 0.05 | 0.01 | -0.06 |
| MIROC-lite | -0.02 | -0.06 | -0.09 |
| P. GENIE | -0.19 | -0.36 | -0.71 |
| UVic | 0.07 | 0.07 | 0.09 |
| Mean | -0.01 | -0.06 | -0.11 |
| Median | -0.01 | -0.05 | -0.08 |
| Standard Deviation | 0.15 | 0.19 | 0.23 |





## 3.2 Effect of emissions rate, 1% vs. Bell

The bell experiments were designed to test whether temperature evolution following cessation of emissions depends on the pathway of emissions before emissions cease. These experiments also illustrate model behaviour during a gradual transition to zero emissions (e.g. MacDougall, 2019), a pathway that is consistent with most future scenarios (Eyring et al., 2016). Nine of the participating models conducted both the A1 and B1 experiments, GFDL and eight of the EMICs (CLIMBER is the EMIC which did not conduct the B1 experiment). Figure 4 shows the temperature evolution (relative to pre-industrial temperature) for both experiments. All models show that by the 100th year of the experiments, when emissions cease in the bell experiment, the temperature evolution is very close in the two experiments. For seven of the models, GFDL, Bern, DCESS, LOVECLIM, MESM, MIROC-lite, and UVic, the temperature evolution in the A1 and B1 experiments is indistinguishable after emissions cease in the Bell experiment. That is, models suggest that in the long term the past pathway of $CO_2$ is largely irrelevant to total temperature change, and determined only by the total amount of cumulative emissions.

Figure 5 shows ZEC for both experiments. There is no sharp discontinuity in forcing in the Bell experiments and thus the temperature of cessation for these experiments is simply calculated relative to a temperature average from a 20 year window centred about the year 100 when emissions cease. Despite the long-term temperature evolution being the same for both experiments, the change in temperature relative to time of cessation is different in most models. This feature is not unexpected as theoretical work on the TCRE relationship suggest that direct proportionality between cumulative emissions of $CO_2$ and temperature change should break down when emission rates are very low (MacDougall, 2017), as emissions are near the end of the Bell experiments. That is, in the type-B experiments emissions decline gradually and hence the Earth system is closer to thermal and carbon cycle equilibrium when emissions cease. These results support using the type-A experiment (1% followed by sudden transition to zero emissions) to calculate ZEC for providing a correction to the remaining carbon budget, as the experiment provides a clear separation between TCRE and ZEC, while for a gradual transition to zero emissions scenario the two effects are mixed as emissions approach zero.

The B2 experiment (750 PgC) was designed to assess ZEC for an emissions total that would imply a climate warming of close to 1.5°C (Jones et al., 2019). The mean change in emission rate for the B2 experiment during the ramp-down phase of the experiment (year 50 to 100) is -0.39 PgC $a^{-2}$. This rate is similar to the rate of -0.29 [-0.05 to -0.64] PgC $a^{-2}$ for stringent mitigation scenarios from the IPCC Special Report on 1.5° for the period from year 2020 to 2050 CE (Rogelj et al., 2018). Therefore we would expect similar behaviour in the stringent mitigation scenarios and the type-B experiments. That is, for the effect of ZEC to manifest while emissions are ramping down. The A2 experiment (1% 750 PgC) branches from the 1pctCO2 experiment between year 51 and 60 in the models that performed that experiment. Emission in the A2 experiment cease in year 100. Thus the temperature correction expected by time emission cease for the stringent mitigation scenarios would be in the range of $ZEC_{40}$ to $ZEC_{50}$ for the A2 experiment.



**Figure 4.** Temperature evolution of A1 (1% to 1000 PgC) and B1 (Bell shaped emissions of 1000 PgC over 100 years) experiments relative to pre-industrial temperature. Solid lines are the A1 experiment and dashed lines are the B1 experiment. Vertical blue line shows when emissions cease in the A1 experiment and vertical red line shows where emissions cease in the B1 experiment.



**Figure 5.** ZEC for the A1 (1% to 1000 PgC) and B1 (Bell shaped emissions of 1000 PgC over 100 years) experiments. Solid lines are the A1 experiment and dashed lines are the B1 experiment.





### 3.3 Sensitivity of ZEC to Cumulative Emissions

Twelve models conducted at least two type-A (1%) experiments such that ZEC could be calculated for 750 PgC, 1000 PgC, and 2000 PgC of cumulative emissions, five ESMs (ACCESS, CanESM5, GFDL, MIROC-ES2L, and UKESM), and all of the EMICs except CLIMBER. Two of the models conducted only two of the type-A experiments, CanESM5 conducted the A1 and A3 experiments, while LOVECLIM conducted the A1 and A2 experiments. Figure 6 shows the $ZEC_{50}$ for each model for the three experiments. All of the full ESMs exhibit higher $ZEC_{50}$ with higher cumulative emissions. The EMICs have a more mixed response with Bern, MESM, LOVECLIM and UVic showing increased $ZEC_{50}$ with higher cumulative emissions, DCESS and IAPRAS showing slightly declining $ZEC_{50}$ with higher cumulative emissions, and P. GENIE showing a strongly declining $ZEC_{50}$ with higher emissions. The inter-model range for the $ZEC_{50}$ of the A2 (750 PgC) experiment is -0.31 to 0.30 °C with a mean value of -0.02°C, a median of -0.04°C and standard deviation of 0.14°C . The inter-model range A3 (2000 PgC) experiment -0.40 to 0.52°C with a mean of 0.12°C, a median of 0.10°C and standard deviation of 0.26°C. Note that different subsets of models conducted each experiment, such that ranges between experiments are not fully comparable.

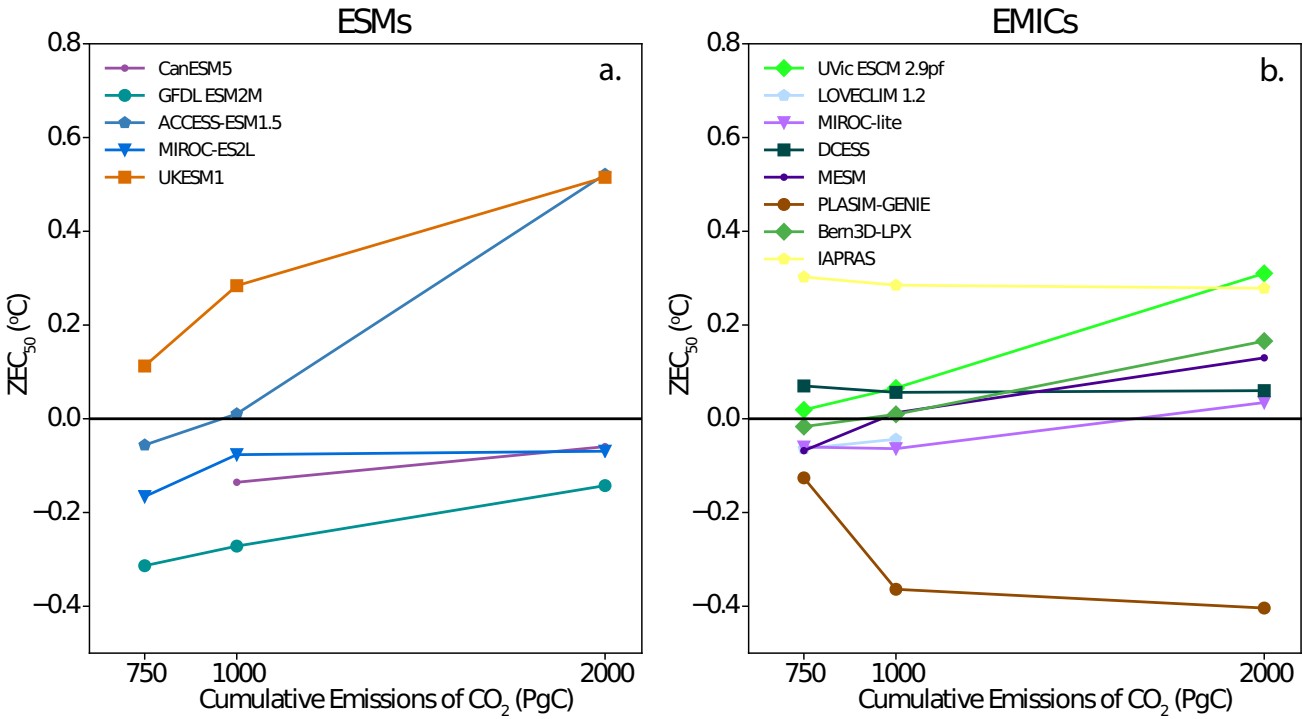

**Figure 6.** Values of $ZEC_{50}$ for the 750, 1000 and 2000 PgC experiments branching from the 1% experiment (type-A). Panel (a) shows results for full ESMs and panel (b) for EMICs



### 3.4 Analysis of Results

The framework introduced in section 2.4 was applied to the ZECMIP output to partition the energy balance components of
ZEC into contributions from the warming effect of the reduction in ocean heat uptake ($-\Delta N$), and the effect of the change in radiative forcing from the ocean and terrestrial carbon fluxes. Figure 7 shows the results of this analysis for each model averaged over the period 40 to 59 years after emissions cease for the 1000 PgC 1% (A1) experiment (the same time interval as $ZEC_{50}$). The results suggest that both ocean carbon uptake and terrestrial carbon uptake are critical for determining the sign of ZEC in the decades following cessation of emissions. Previous efforts to examine ZEC, while acknowledging the terrestrial
carbon sink, have emphasized the role of ocean heat and carbon uptake (Ehlert and Zickfeld, 2017; Williams et al., 2017). These studies also focused on ZEC on timescales of centuries, not decades. In CanESM5 and CNRM the terrestrial carbon sink dominates the reduction in radiative forcing, while in ACCESS, IAPRAS, MESM, P. GENIE, and UVic the ocean carbon uptake dominates the reduction in radiative forcing. The remaining models have substantial contributions from both carbon sinks. In all models the reduction in forcing from ocean carbon uptake is smaller than the reduction in ocean heat uptake, suggesting
that the post-cessation net land carbon sink is critical to determining ZEC values. Given that the behaviour of the terrestrial carbon cycle varies strongly between models (Friedlingstein et al., 2006; Arora et al., 2013, 2019) and that many models lack feedbacks such as nutrient limitation and permafrost carbon pools, the strong dependence of $ZEC_{50}$ on terrestrial uptake is concerning. Notably the three ESMs with the weakest modelled terrestrial carbon sink response (ACCESS, MIROC-ES2L, and UKESM) are three which include terrestrial nutrient limitations (Table 2, 3). The UVic model includes permafrost carbon
and has a relatively weak terrestrial carbon uptake (Table 5). However, Bern and MPI-ESM also have nutrient limitations and have a terrestrial carbon uptake in the middle (Bern) and upper (MPI-ESM) parts of the inter-model range. IAPRAS does not account for either nutrient limitations or permafrost carbon and has the weakest terrestrial carbon uptake of all (Table 4). Ocean carbon uptake also varies substantially between models, with some of the EMICs (P. GENIE, MESM, and IAPRAS) having very high ocean carbon uptake, and two of the ESMs (CanESM5 and CNRM) and having very low ocean carbon uptake.

Figure 8 shows the relationship between ocean heat uptake, cumulative ocean carbon uptake, and the cumulative terrestrial carbon uptake when emissions cease and 50 years after emissions cease. Excluding the clear outlier of IAPRAS Figure 8 shows a clear negative relationship (R=-0.87) between ocean heat uptake before emissions cease and the change in ocean heat uptake 50 years after emissions cease in the A1 experiment. That is, models with high ocean uptake before emissions cease tend to have a strong reduction in ocean heat uptake after emissions cease. Similarly there is a strong (R=0.88) positive relationship
between ocean carbon uptake before emissions cease and uptake in the 50 years after emissions cease. The relationship between uptake (or in one case net release) of carbon by the terrestrial biosphere before and after emissions cease is weaker (R=0.64) but clear. Therefore explaining why the energy balance components illustrated by Figure 7 vary between models would seem to relate strongly to why models have varying ocean heat, ocean carbon uptake, and terrestrial carbon cycle behaviour before emissions cease.

It has long been suggested that the reason that long-term ZEC was close to zero is compensation between ocean heat and ocean carbon uptake (Matthews and Caldeira, 2008; Solomon et al., 2009; Frölicher and Paynter, 2015), which are both partially





controlled by deep ocean circulation (Banks and Gregory, 2006; Xie and Vallis, 2012; Frölicher et al., 2015). However, Figure 7 shows that this generalization is not true for decadal time-scales. The two quantities do compensate one-another but in general the effect from reduction in ocean heat uptake is larger than the change in radiative forcing from the continued ocean carbon

uptake. Thus going forward additional emphasis should be placed on examining the role of the terrestrial carbon sink in ZEC for policy relevant timescales.

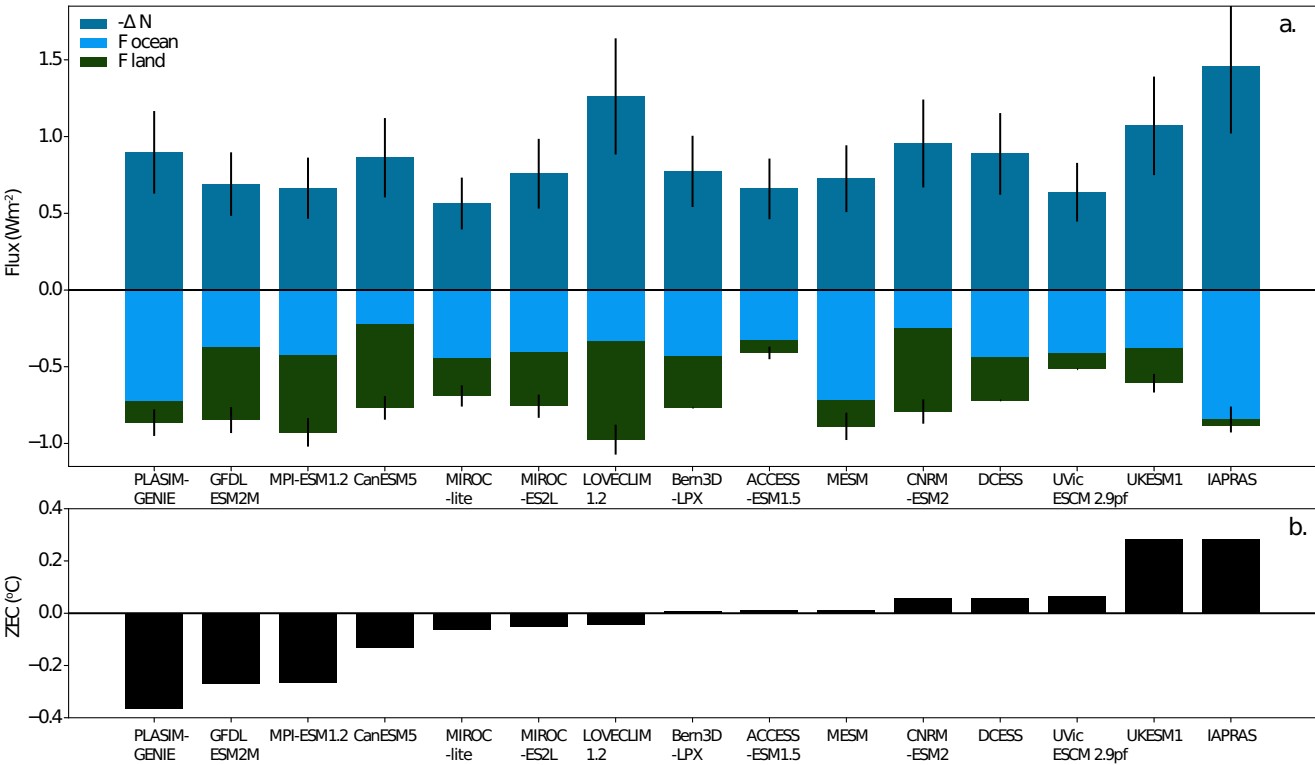

**Figure 7.** (a) Energy fluxes following cessation of $CO_2$ emissions for the 1000 PgC 1% (A1) experiment. $\Delta N$ is the change in ocean heat uptake relative to the time emissions ceased. A reduction in ocean heat uptake will cause climate warming, hence $-\Delta N$ is displayed. $F_{ocean}$ is the change in radiative forcing caused by ocean carbon uptake, and $F_{land}$ is the change in radiative forcing caused by terrestrial carbon uptake. Vertical black lines are estimated uncertainty ranges. (b) $ZEC_{50}$ values for each model. Models are arranged in ascending order of $ZEC_{50}$

Figure 9 compares the energy fluxes for the ten models that conducted all of the type-A (1%) experiments. All three energy balance components seem to be affected by the cumulative emissions leading up to cessation of emissions, however there is no universal pattern. Most models show a larger reduction in ocean heat uptake with higher cumulative emissions, but UKESM

has the largest reduction for the 1000 PgC experiment. Variations in the reduction in radiative forcing from ocean carbon uptake tend to be small between simulations within each model but show no consistent patterns between models. Most of the models show a smaller terrestrial carbon sink for the 2000 PgC experiment than the other two experiments, the exception being



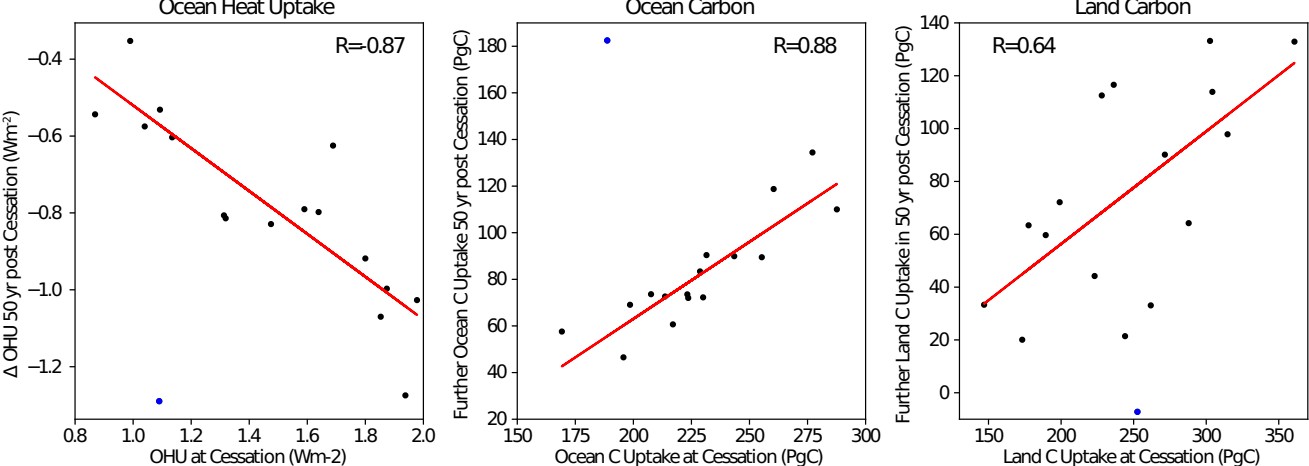

**Figure 8.** Relationship between variables before emissions cease and 50 years after emissions cease. (a) Ocean Heat Uptake (OHU) is computed for 20 year windows with the value at cessation taken from the 1pctCO2 experiment analogous to the how temperature of cessation is computed. (b) Cumulative ocean carbon uptake, (c) Cumulative land carbon uptake. Each marker represents value from a single model. Line of best fit excludes the outlier model IAPRAS which is marked in blue.

IAPRAS which shows the opposite pattern. Examining in detail why these factors change in each model could be a productive avenue for future research.

## 3.5 Relationship to other climate metrics

Figure 10 shows the relationship between ECS, TCR, TCRE, Realized Warming and $ZEC_{50}$ for the A1 (1000 PgC) experiment. Realized warming is the ratio of TCR to ECS. TCR is transient warming when $CO_2$ is doubled and ECS is warming at equilibrium following doubling of $CO_2$, their ratio is the fraction of warming from $CO_2$ that has been realized, hence 'Realized Warming' (e.g. Frölicher et al., 2014). ECS shows virtually no correction to $ZEC_{50}$ (R=0.03) and thus ZEC and ECS appear to be independent. Both TCR and Realized Warming show weak positive correlations with $ZEC_{50}$ (R= 0.23 and 0.29 respectively). TCRE shows the strongest relationship to $ZEC_{50}$ with a correlation coefficient of 0.45. However the relationship is weak and several models with high TCRE values have low ZEC values. The poor correlation between ZEC and other climate metrics is not unexpected as ZEC is determined by the difference in warming caused by reduction in ocean heat uptake and cooling caused by continued land and ocean carbon uptake after the cessation of emissions. Small differences between large quantities are not expected to correlated well to the quantities used to calculate them.

Bern and UVic both conducted the ZECMIP experiments with three versions of their models with different equilibrium climate sensitivities, allowing for examination of the effect of ECS on ZEC. Figure 11 shows the $ZEC_{50}$ for these simulations. The figure shows that for both Bern and UVic higher ECS corresponds to higher ZEC. For Bern for the A1 (1% 1000 PgC) experiment $ZEC_{50}$ is 0.01, 0.03, and 0.18°C for ECSs of 2.0, 3.0 and 5.0 °C respectively. For UVic for the A1 (1% 1000 PgC) experiment $ZEC_{50}$ is 0.01, 0.07, and 0.32 for ECSs of 2.0, 3.5 and 4.5 °C respectively. Note that the ECS values given here





**Figure 9.** Energy fluxes following cessation of $CO_2$ emissions for the type-A experiments (1%) for each model. $\Delta N$ is the change in ocean heat uptake relative to the time emissions ceased. A reduction in ocean heat uptake will cause climate warming, hence $-\Delta N$ is displayed. $F_{ocean}$ is the change in radiative forcing caused by ocean carbon uptake, and $F_{land}$ is the change in radiative forcing caused by terrestrial carbon uptake. All fluxes are computed for averages from 40 to 59 years after emissions cease.

are for true equilibrium climate sensitivity, not Gregory ECS as used in the remainder of this study. Figure 12 compares the energy fluxes for the three versions of Bern and UVic. For both Bern and UVic ocean carbon uptake is unaffected by climate sensitivity. For Bern the reduction in ocean heat uptake is larger at higher climate sensitivity, while for UVic this quantity is almost the same for ECSs of 3.5 and 4.5 °C. In both models the terrestrial carbon sink is weaker in versions with higher climate sensitivity. Overall the results suggest a relationship between higher ECS and higher ZEC within these models.



**Figure 10.** Relationship between: ECS (a), TCR (b), TCRE (c), Realized Warming (d) and $ZEC_{50}$. Line of best fit is shown in red. Correlation coefficients are displayed in each panel.





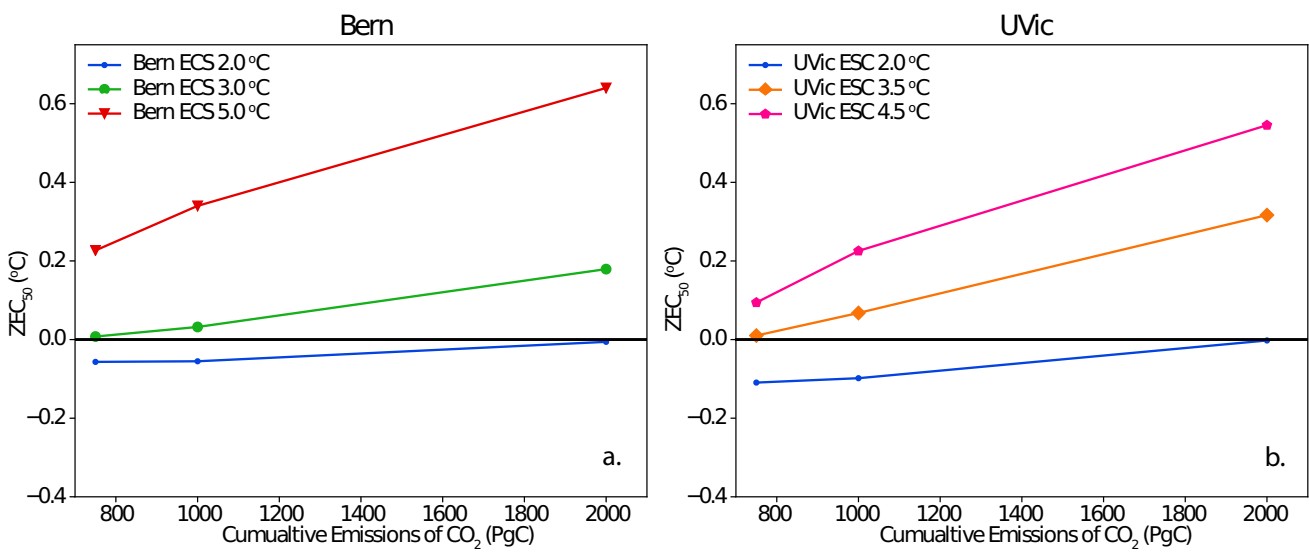

**Figure 11.** Values of 50 year ZEC for the 750, 1000 and 2000 PgC experiments branching from the type-A (1%), for versions of Bern and UVic with varying Equilibrium Climate Sensitivity. Note that the ECS values given here are for true equilibrium climate sensitivity, not Gregory ECS as used in the remainder of the study.

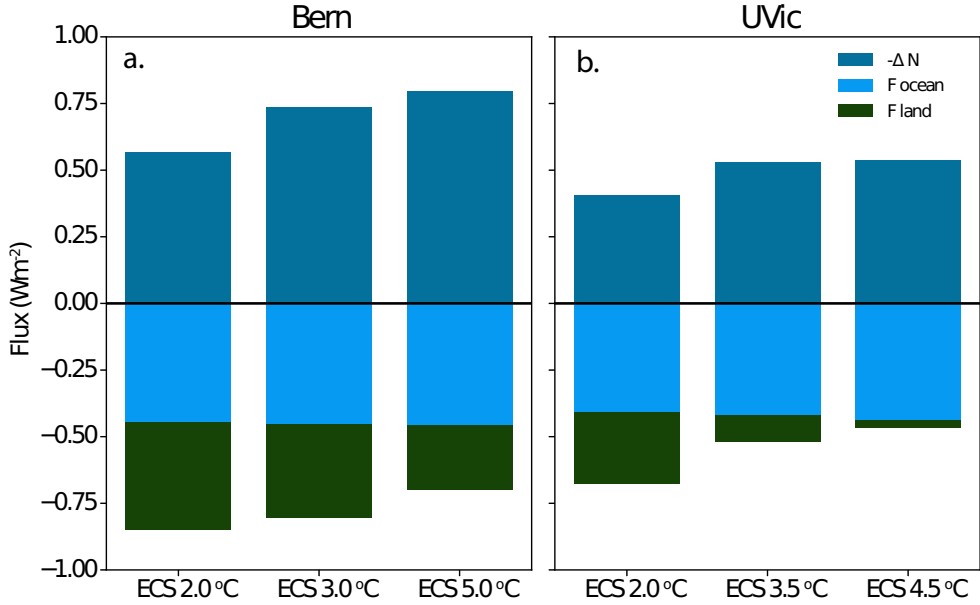

**Figure 12.** Energy fluxes following cessation of $CO_2$ emissions for the type-A experiments (1%) for versions of Bern and UVic with varying Equilibrium Climate Sensitivity. $\Delta N$ is the change in ocean heat uptake relative to the time emissions ceased. A reduction in ocean heat uptake will cause climate warming, hence $-\Delta N$ is displayed. $F_{ocean}$ is the change in radiative forcing caused by ocean carbon uptake, and $F_{land}$ is the change in radiative forcing caused by terrestrial carbon uptake. All fluxes are computed for averaged from 40 to 59 years after emissions cease.





## 4 Discussion

### 4.1 Drivers of ZEC

The analysis here has shown that ZEC is poorly correlated to other metrics of climate warming, such as TCR and ECS.
However, the three factors that drive ZEC, ocean heat uptake, ocean carbon uptake and net land carbon flux correlate relatively

well to their states before emissions cease. Thus it may be useful to conceptualize ZEC as a function of these three components
each evolving in their own way in reaction to the cessation of emissions. Ocean heat uptake evolves due to changes in ocean
dynamics (e.g. Frölicher et al., 2015) as well as the complex feedbacks that give rise to changes in ocean heat uptake efficacy
(Winton et al., 2010). Ocean carbon uptake evolution is affected by ocean dynamics, changes to ocean biogeochemistry, and
changes in atmosphere-ocean $CO_2$ chemical disequilibrium, where the latter is also influenced by land carbon fluxes (e.g.

Frölicher et al., 2015). The response of the land biosphere to cessation of emissions is expected to be complex with contributions
from, the response of photosynthesis to declining atmospheric $CO_2$ concentration, a continuation of enhanced soil respiration
(e.g. Jenkinson et al., 1991) and release of carbon from permafrost soils (Schuur et al., 2015), among other factors. Investigating
the evolution of the three components in detail may be a valuable avenue of future analysis. Similarly, given their clearer
relationships to the state of the Earth system before emissions cease, focusing on the three components independently may

prove useful for building a framework to place emergent constraints on ZEC. Future work will explore evaluation opportunities
by assessing relationships between these quantities in the idealized 1% simulation and values at the end of the historical
simulations up to present day.

### 4.2 Policy Implications

One of the main motivations to explore ZEC are its implications for policy and society's ability to limit global warming to

acceptable levels. Climate policy is currently aiming at limiting global mean temperature increase to well below 2 °C and
pursuing to limit it to 1.5 °C (United Nations, 2015). To stay within these temperature limits, emission reduction targets are
being put forward. These targets can take the form of emissions reductions in specific years, like the Paris Agreement nationally
determined contributions for the years 2025 or 2030 (Rogelj et al., 2016), but also of net zero emissions targets that cap the
cumulative $CO_2$ emissions a country is contributing to the atmosphere (Haites et al., 2013; Rogelj et al., 2015; Geden, 2016).

Because ZEC does affect the required stringency of emissions reductions or of the maximum warming one can expect, it is
important to clearly understand its implications within a wider policy context. First, for policy analysts and scientists, the
quantification of $ZEC_{50}$ will help inform better estimates of the remaining carbon budget compatible with limiting warming
to 1.5 °C or well below 2 °C over the course of this century. Analysts need to be clear, however, that $ZEC_{50}$ is only then an
adequate adjustment for TCRE-based carbon budget estimates if the TCRE values are based on 1% $CO_2$ increase simulations.

In contrast, however, our results also show that when $CO_2$ emissions ramp down gradually (c.f. the B-series of ZECMIP
experiments) $ZEC_{50}$ is generally much smaller, because part of it is already realized during the emissions ramp down. Hence
this means that in a situation in which society successfully gradually reduces its global $CO_2$ emissions to net zero at rates
comparable to the B2 experiment (see Section 3.2), the expected additional warming on time scales of decades to maximum a



century is small. Finally, over multiple centuries, warming might still further increase or decrease. In the former case, a certain

level of carbon-dioxide removal will be required over the coming centuries. The level implied by the long-term ZEC, however, represents much less a challenge than the urgent drastic emissions cuts required to limit warming to either 1.5 °C or 2 °C over the next decades (Rogelj et al., 2019b).

## 4.3 Moving towards ZECMIP-II

For the first iteration of ZECMIP the experimental protocol has focused solely on the response of the Earth system to zero

emission of $CO_2$. However, many other non-$CO_2$ greenhouse gases, aerosol, and land-use-changes all affect global climate (e.g. IPCC, 2013). To truly explore the question whether global temperature will continue to increase following complete cessation of greenhouse gas and aerosol emissions the effect of each anthropogenic forcing agent must be accounted for (e.g. Frölicher and Joos, 2010; Matthews and Zickfeld, 2012; Mauritsen and Pincus, 2017; Allen et al., 2018; Smith et al., 2019). We envision a second iteration of ZECMIP accounting for these effects with a set of self-consistent idealized experiments, as

part of the formal CMIP7 process.



## 5 Conclusions

Here we have analyzed model output from the 18 models that participated in ZECMIP. We have found that the inter-model range of ZEC 50 years after emissions cease for the A1 (1% 1000 PgC) experiment is -0.36 to 0.29 °C with a model ensemble mean of -0.06°C, median of -0.05°C, and standard deviation of 0.19°C. Models show a range of temperature evolution after
emissions cease from continued warming for centuries to substantial cooling. All models agree that following cessation of $CO_2$ emissions, the atmospheric $CO_2$ concentration will decline. Comparison between experiments with a sudden cessation of emissions and a gradual reduction in emissions show that long term temperature change is independent of the pathway of emissions. However, in experiments with a gradual reduction in emissions the temperature adjustment from the values expected from TCRE (i.e. ZEC) begin to manifest while emissions are ramping-down. As the rate of emission reduction in
these idealized experiments is similar to that in stringent mitigation scenarios, a similar pattern may emerge if deep emission cuts commence.

ESM simulations agree that higher cumulative emissions lead to a higher ZEC, though some EMICs show the opposite relationship. Analysis of the model output shows that both ocean carbon uptake and the terrestrial carbon uptake are critical for reducing atmospheric $CO_2$ concentration following the cessation of $CO_2$, and thus counteracting the warming effect of
reduction in ocean heat uptake. ZEC is poorly correlated to other Earth system metrics, however, the three factors that contribute to ZEC (ocean heat uptake, ocean carbon uptake and net land carbon flux) correlate well to their states prior to the cessation of emissions.

Overall, the most likely value of ZEC on decadal time-scales is assessed to be close to zero, consistent with prior work. However substantial continued warming for decades or centuries following cessation of emission is a feature of a minority of
the assessed models and thus cannot be ruled out.



*Data availability.* ESM data will be published and freely available as per CMIP6 data policy on the Earth System Grid Federation (https://esgf-node.llnl.gov/projects/cmip6/ ). EMIC data will be published and freely available on a dedicated server (terra.seos.uvic.ca/ZEC). The annual global mean variables used for the present analysis will also be made available on the server.





**Appendix A: Analytical framework**

A key question for this study is explaining why some models have positive ZECs and some models have negative or close to zero ZECs. From elementary theory we understand that the sign of ZEC will depend on the pathway of atmospheric $CO_2$ concentration and ocean heat uptake following cessation of emissions. Complicating this dynamic is that atmospheric $CO_2$ change has contributions both from ocean carbon uptake and the net flux from the terrestrial biosphere. Here we devise a simple method for partitioning the contribution to ZEC from the ocean carbon flux, net land carbon flux, and the ocean heat

uptake.

We begin with the forcing response equation (Wigley and Schlesinger, 1985):

$$F = \lambda T + \epsilon N, \tag{A1}$$

where $F$ (W m$^{-2}$) is radiative forcing, $N$ (W m$^{-2}$) is planetary heat uptake, $\epsilon$ (dimensionless) is the efficacy of planetary heat uptake, $\lambda$ (W m$^{-2}$K$^{-1}$) is the climate feedback parameter, and $T$ (K) is the change in global temperature (relative to

pre-industrial). This equation can be re-written as:

$$\lambda T = F - \epsilon N. \tag{A2}$$

To compute the rate of change of temperature we take the derivative of equation A2 in time giving:

$$\lambda \frac{dT}{dt} = \frac{dF}{dt} - \epsilon \frac{dN}{dt}. \tag{A3}$$

Radiative forcing from $CO_2$ can be approximated using the classical logarithmic relationship (Myhre et al., 1998):

$$F = R \ln \left( \frac{C_A}{C_{Ao}} \right), \tag{A4}$$

where $R$ (W m$^{-2}$) is the radiative forcing from an e-fold increase in atmospheric $CO_2$ concentration, $C_A$ (PgC) is atmospheric $CO_2$ burden and $C_{Ao}$ (PgC) is the original atmospheric $CO_2$ burden. Recalling that the derivative of $\frac{d\ln(x)}{dx} = \frac{1}{x}$ the derivative of equation A4 is:

$$\frac{dF}{dt} = R \left( \frac{C_{Ao}}{C_A} \right) \left( \frac{dC_A}{dt} \right) \frac{1}{C_{Ao}}, \tag{A5}$$

which simplifies to:

$$\frac{dF}{dt} = \frac{R}{C_A} \left( \frac{dC_A}{dt} \right). \tag{A6}$$





After emissions cease atmospheric $CO_2$ concentration can be expressed as:

$$C_A = C_{ze} - (C_O - C_{Oze}) - (C_L - C_{Lze}),$$ (A7)

Where $C_{ze}$ (PgC) is atmospheric $CO_2$ burden at the time emissions reach zero, $C_O$ (PgC) is the carbon content of the ocean, and $C_L$ (PgC) is the carbon content of land. $C_{Oze}$ (PgC) is the carbon content of the ocean at the time emissions reach zero and $C_{Lze}$ (PgC) is the carbon content of land when emissions reach zero. Thus the derivative of $Ca$ is:

$$\frac{dC_A}{dt} = -\frac{dC_O}{dt} - \frac{dC_L}{dt}$$ (A8)

$\frac{dC_O}{dt}$ is the flux of carbon into the ocean $f_O$, and $\frac{dC_L}{dt}$ is the flux of carbon into land $f_L$.

$$\frac{dC_A}{dt} = -f_O - f_L.$$ (A9)

Substituting equation A9 in equation A6 we find:

$$\frac{dF}{dt} = -R\frac{f_O + f_L}{C_A},$$ (A10)

which can be split into:

$$\frac{dF}{dt} = -R\frac{f_O}{Ca} - R\frac{f_L}{C_A},$$ (A11)

which can be substituted into equation A3:

$$\lambda\frac{dT}{dt} = -R\frac{f_O}{C_A} - R\frac{f_L}{C_A} - \epsilon\frac{dN}{dt}.$$ (A12)

If we integrate equation A12 from time emissions reach zero we get:

$$\lambda T_{ZEC} = -R\int_{t=ze}^{\infty}\frac{f_O}{Ca}dt - R\int_{t=ze}^{\infty}\frac{f_L}{C_A}dt - \epsilon(N - N_{ze}),$$ (A13)

where $T_{ZEC}$ (K) is ZEC, $N_{ze}$ (W m$^{-2}$) is the planetary heat uptake when emissions cease. The integrals $\int_{t=ze}^{\infty}\frac{f_O}{C_A}dt$ and $\int_{t=ze}^{\infty}\frac{f_L}{C_A}dt$ can be computed numerically from ZECMIP output. Therefore we define:

$$F_{ocean} = R\int_{t=ze}^{\infty}\frac{f_O}{C_A}dt,$$ (A14)





and,

$$F_{land} = R \int\limits_{t=ze}^{\infty} \frac{f_L}{C_A} dt;$$  (A15)

and thus:

$$\lambda T_{ZEC} = -F_{ocean} - F_{land} - \epsilon(N - N_{ze}),$$  (A16)





## 480 Appendix B: ZEC values for A2 and A3 experiments

**Table A1.** Temperature anomaly relative to the year emissions cease averaged over a 20 year time window centred about the 25th, 50th, and 90th year following cessation of anthropogenic $CO_2$ emissions ($ZEC_{25}$, $ZEC_{50}$, and $ZEC_{90}$ respectively) for the A2 (750 PgC 1% experiment).

| Model | $ZEC_{25}$ (°C) | $ZEC_{50}$ (°C) | $ZEC_{90}$ (°C) |
|---|---|---|---|
| ACCESS | -0.03 | -0.06 | -0.12 |
| GFDL | -0.26 | -0.31 | -0.26 |
| MIROC-ES2L | -0.04 | -0.17 | -0.20 |
| UKESM | 0.13 | 0.11 | 0.08 |
| Bern | 0.03 | -0.02 | -0.09 |
| DCESS | 0.12 | 0.07 | -0.02 |
| IAPRAS | 0.34 | 0.30 | 0.08 |
| LOVECLIM | -0.06 | -0.06 | 0.05 |
| MESM | 0.0 | -0.07 | -0.13 |
| MIROC-lite | 0.0 | -0.06 | -0.09 |
| P. GENIE | 0.03 | -0.13 | -0.12 |
| UVic | 0.06 | 0.02 | 0.06 |
| Mean | 0.03 | -0.02 | -0.06 |
| Median | 0.02 | -0.04 | -0.09 |
| Standard Deviation | 0.13 | 0.14 | 0.10 |

**Table B1.** Temperature anomaly relative to the year emissions cease averaged over a 20 year time window centred about the 25th, 50th, and 90th year following cessation of anthropogenic $CO_2$ emissions ($ZEC_{25}$, $ZEC_{50}$, and $ZEC_{90}$ respectively) for the A3 (2000 PgC 1% experiment).

| Model | $ZEC_{25}$ (°C) | $ZEC_{50}$ (°C) | $ZEC_{90}$ (°C) |
|---|---|---|---|
| ACCESS | 0.21 | 0.52 | 0.65 |
| CanESM5 | 0.01 | -0.06 | -0.05 |
| GFDL | -0.12 | -0.14 | -0.06 |
| MIROC-ES2L | 0.03 | -0.07 | -0.12 |
| UKESM | 0.37 | 0.51 | 0.75 |
| Bern | 0.17 | 0.17 | 0.15 |
| DCESS | 0.09 | 0.06 | -0.01 |
| IAPRAS | 0.34 | 0.28 | 0.0 |
| MESM | 0.12 | 0.13 | 0.11 |
| MIROC-lite | 0.08 | 0.03 | -0.02 |
| P. GENIE | -0.15 | -0.40 | -0.48 |
| UVic | 0.23 | 0.31 | 0.44 |
| Mean | 0.12 | 0.11 | 0.11 |
| Median | 0.11 | 0.10 | -0.01 |
| Standard Deviation | 0.16 | 0.26 | 0.33 |





## Appendix C: Efficacy Evolution

**Figure A1.** Evolution of efficacy for the four EMICs without substantial internal variability. Red horizontal line is the efficacy value estimated from the 1pctCO2 experiment and vertical blue line marks 50 years after emissions cease.

*Author contributions.* CDJ, TLF, AHMD, JR, HDM, and KZ initiated the research. CDJ and AHMD coordinated the project. AHMD, VKA, VB, FAB, AVE, TH, PBH, AJT, CK, LM, MM, IIM, RS, GS, JT, JS,TZ, KT, and AO carried out model the simulations. AHMD, CDJ, TLF, JR, HDM, and KZ contributed to the analysis. ME and NJB contributed critical technical support and programming. AHMD, HDM, VKA, and JR contributed significantly to the writing of the paper.

*Competing interests.* AVE is a member of the journal Editorial Board. All other authors declare no competing interests





*Acknowledgements.* AHMD, ME, HDM, and KZ are each grateful for support from respective National Science and Engineering Research Council of Canada Discovery Grants. TLF and FAB acknowledge support from the Swiss National Science Foundation under grant PP00P2 - 170687 and from the European Union's Horizon 2020 Research and Innovative Programme under grant agreement number 821003 (CCiCC).

GFDL ESM2M simulations were performed at the Swiss National Supercomputing Centre (CSCS). CDJ and AW acknowledge the Joint UK BEIS/Defra Met Office Hadley Centre Climate Programme (GA01101) and the EU Horizon 2020 CRESCENDO project, grant number 641816. JR, RS and MM acknowledge support from the European Union's Horizon 2020 Research and Innovation Programme under grant agreement number 820829 (CONSTRAIN). VB is grateful to Thomas Raddatz and Veronika Gayler for assistance with the MPI-ESM experimental setup and acknowledges support from the EU Horizon 2020 CRESCENDO project. AVE was supported by the Russian Foundation

for Basic Research (grant 18-05-00087). TH and KT acknowledge the support from "Integrated Research Program for Advancing Climate Models" by the Ministry of Education, Culture, Sports, Science, and Technology of Japan. PBH was funded by LC3M, a Leverhulme Trust Research Centre Award. AJT acknowledges support by the Swiss National Science Foundation (# 200020_172476) and the Oeschger Centre for Climate Change Research. IIM was supported by the Russian Science Foundation (grant 19-17-00240). GS was supported by FONDE-CYT (Chile) grant 1190230. LM acknowledges support from the Australian Research Council FT180100606. RS and MM acknowledge the

EU Horizon 2020 CRESCENDO project, grant number 641816 and Supercomputing time provided by the Météo-France/DSI supercomputing center. JT and JS acknowledge Research Council of Norway funded project KeyClim (295046).

The authors thank all participants of the expert workshop on carbon budgets, co-organized with the support of the Global Carbon Project, the CRESCENDO project, the University of Melbourne, and Simon Fraser University, for discussions that initiated the authors' thinking about ZECMIP.



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
