# Peer review of "Is there warming in the pipeline? A multi-model analysis of the zero emission commitment from $CO_2$"

_Biogeosciences, 2019_

## Referee Comment (RC1) · Anonymous Referee #1 · 24 Feb 2020

Review of "Is there warming in the pipeline? A multi..." by A. H. MacDougall and co-authors.

In this model intercomparison the global mean temperature response following a ceasure of CO2 emissions is investigated. This is a welcome addition to a literature mostly based on disparate evidence from single models. The study is mostly well conducted, and can in my opinion be published after the below mostly minor issues have been adressed.

Suggestions:

14, I would add "and simple theory"

[Figure]

150, I would delete ", often called ...". I personally have never heard the expression "Gregory ECS"

153, Andrews et al. (2012) use years 1 to 150, not 20 to 140 as stated here

171-172, if the method is unbiased, how come nearly all models have negative ZEC at year zero in Figure 2b?

210, the way efficacy, or epsilon, is calculated it folds in state-dependence of feedbacks into the epsilon. Probably not an essential issue in this paper, since this parameter isn't important for ZEC, though worth a few words of mention

210, on the same theme, the study would benefit from a more up-to-date treatment of efficacy. In newer studies this is referred to as time-dependent feedback or pattern-effects.

218, delete 'of the models'

227-229, this sentence is confusing. Models may in addition exhibit cloud adjustments not taken into account in neither Myhre nor Etminan. Therefore models may well lay outside (and they do). I recommend deleting.

239-240, somewhat repetitive.

241, I believe the authors have mistaken MPI-ESM for Loveclim.

249-250, any reason we should pay attention to such awkward behaviour?

299-344, here I didn't understand why the authors refer to a diagnostic framework (Equation 4), and then don't use it? In particular if ZEC is non-zero, then climate change feedbacks (lambda) will affect the balance, and likewise if there is ocean heat uptake efficacy (non-unitary epsilon).

319, delete last instance of 'and'

Figure 8, I recommend using a different colour than blue for the outlier, in print it was

[Figure]

hard to distinguish.

346-355, here I would like to see a statistical test that the slopes are actually non-zero. To me it seems rather random, for instance the slope in panel c would probably disappear if omitting the single high TCRE model.

361, use another word than 'Gregory ECS'

368, any reason to think that ZEC should be related to these quantities? In fact, I think it is great that it is not, since basic theory suggests it shouldn't.

400, the phrasing 'will be required', seems to suggest that sustaining a constant global temperature somehow is desirable. I suggest leaving it to others to decide this.

418-419, I didn't understand this sentence.

425, since we don't expect ZEC to be related to ECS, TCR and TCRE, why start the sentence like this?

430, I recommend amending: '... cannot be ruled out purely on the basis of models.'

---

## Referee Comment (RC2) · Ric Williams (Referee) · 28 Feb 2020

The study is addressing an important unknown in the climate system as to whether there is continued warming or cooling on a decadal timescale after carbon emissions cease. There is a comprehensive analysis of 18 Earth system models (of either full or intermediate complexity) following a common default experiment of CO2 increasing until 1000 PgC has been emitted and then freely varying.

I am very positive towards the ambition, scope and rigour of the manuscript. A key outcome is the message is that ocean and terrestrial carbon uptake is particularly important in determining the Zero Emissions Commitment (ZEC). I have 2 concerns

with regards the strength of this conclusion.

1. The analysis of the thermal response on a decadal timescale focuses on changes in planetary heat uptake and the efficacy, representing a non-dimensional weighting of planetary heat uptake. The study ignores any explicit discussion of temporal changes in climate feedback parameter (due to their method to diagnose the effective climate sensitivity), particularly associated with clouds. Since the study is focussing on decadal timescales, this omission is likely to be important and should be more fully discussed. Clouds represent one of the biggest uncertainties in climate sensitivity and are known to evolve in time. This potentially important contribution of physical climate feedbacks merits a fuller discussion.

I fully realise that time-variations of the climate feedback parameter may be related to temporal variability in the efficacy. Indeed the authors include a 30% uncertainty in the planetary heat uptake term to take account of this effect, but do not extensively discuss the implications of this uncertainty. In their crucial figure 7, the sign of the ZEC in most model cases is uncertain given the uncertainty arising from the efficacy weighting of the change in the planetary heat uptake. Thus, there needs to be more acknowledgement of the uncertainty in the ZEC due to physical climate feedbacks (or an explanation of why these physical climate feedbacks are less uncertain at the time of zero emission). Whether this uncertainty represents random variability or a systematic trend needs to be addressed.

2. There is a larger inter-model spread in the response of the terrestrial carbon feedback. A key conclusion is that the terrestrial carbon response is of central importance in dictating whether surface temperature continues to rise or fall after emissions ceases. However, this conclusion needs to be seriously caveated by the choice of terrestrial carbon cycle and whether nutrient limitation is included. It might be the case that the terrestrial carbon cycle is becoming over strong if there are no constraining limitations applied. It would be useful to group the analysis of the terrestrial response into those model responses with and without nutrient limitation, and then more clearly contrast

their behaviour.

A further recommendation is in the final conclusion is to recap as to how this work compares with prior studies, particularly for the multi-centennial timescale. This context is set out earlier in the motivation, but it is unclear as to the extent of agreement or not with the inferences in the prior work.

In summary, I think that the study is important and recommend minor edits, particularly to discuss the outcomes of the study in terms of the effect of physical climate feedbacks (via the efficacy) and the controls of nutrient limitation in the terrestrial system, and placing the study in the context of prior work.

Detailed points:

L12 Mention the large uncertainty in the sign of the ZEC from the 30% uncertainty in the efficacy.

L71/72. Both prior studies are addressing the multi-centennial timescale, rather than the previous discussion of a millennial timescale (L30-41). Your study should relate back to these two prior studies and identify what is different to the arguments outlined by Ehlert and Zickfeld (2017) and Williams et al. (2017).

L77 Recommend rephrase to avoid ambiguity so as to make clear to the reader what part of the sentence "only" refers to.

P8-11 Recommend placing the model descriptions in Tables 2 to 5 in the Appendix.

L175. Add Williams et al. (2017) for the multi-centennial case as that study has addressed the different thermal and carbon controls for delayed warming.

P13. In contrast to the model descriptions, I think that the theory in Appendix A could have been placed in the main text, but up to the authors discretion.

L198 include that the change in ocean heat uptake includes a time-dependent weighting from the efficacy.

L209. Include that the Gregory ECS is a time average fit over the time period of interest, while the efficacy is time dependent.

L217 The efficacy values may not simply be representative of internal variability, but may also be associated with systematic shifts in climate feedback, such as systematic changes in cloud types with changing surface temperature.

L223 The authors have acknowledged the importance of the efficacy by including a 30% uncertainty. However, is there a systematic trend to the efficacy or are there random variations in the different models? In diagnostics of ESM2M, there is a progressive increase in the efficacy from close to 1.5 to over 2 in 100 years after emissions cease, which may be equivalently interpreted as a systematic decrease in climate feedback parameter that continues for several centuries; see Williams et al. (2017). If there is a systematic trend, then the implications for the ZEC are different to if there is simply uncertainty in how the efficacy evolves.

L254. Clarify the timescale.

Figure 2. It is difficult to pick out individual model types, particularly those coloured blue and green. Recommend split the panels and show the responses for different types of models to gain more insight.

L333 The response is interpreted in terms of changes in the "deep ocean circulation". However, the anthropogenic invasion of heat and carbon is being dominated by ventilation of the thermocline, see Sabine et al. (2004) Science or Zanna et al. (2019) PNAS. Schematic figure of Goodwin et al. (2015) Nature Geosciences or model diagnostics in Williams et al. (2017) J. Climate set out this thermocline ventilation view.

L335. The large uncertainty in the efficacy weighting of ocean heat uptake in Figure 7 can change the sign of the ZEC and so this aspect is certainly of comparable importance to the terrestrial carbon sink.

Figure 10. It is striking how low a proportion of the intermodel variability in the ZEC

is explained by any of these metrics. Do these fits improve or alter if different subsets of models are included? The relatively weak fits suggests that the ZEC is being determined by a competition of processes and examining one process in isolation only provides limited insight.

Figure 11 is more encouraging in showing that for the same model type, there is a relationship between the ZEC and the TCRE, with an increasing ZEC with a higher ECS. L368 There is a poor relationship between the ZEC and the TCR and ECS when looking across a range of models, but there is a stronger relationship when looking at the same model.

L399 Mention that over multiple centuries that warming might further increase or decline. Useful to expand upon that statement and compare further with the prior studies examining the multi-centennial response.

---

## Author Comment (AC1) · 28 Apr 2020

**Response to reviews of manuscript:**

Is there warming in the pipeline? A multi-model analysis of the zero emission commitment from $CO_2$

 We appreciate the thoughtful comments of both reviewers and have responded to each comment below. The reviews are copied verbatim and are *italicized*. Author responses are in regular font. Changes made to the manuscript are blue.

**Reviewer 1:**

*Review of "Is there warming in the pipeline? A multi..." by A. H. MacDougall and coauthors. In this model intercomparison the global mean temperature response following a ceasure of CO2 emissions is investigated. This is a welcome addition to a literature mostly based on disparate evidence from single models. The study is mostly well conducted, and can in my opinion be published after the below mostly minor issues have been addressed.*

Many thanks for the positive and encouraging review.

*Suggestions:*
*14, I would add "and simple theory"*

The change has been made.

*150, I would delete ", often called ...". I personally have never heard the expression "Gregory ECS"*

Gregory ECS has been changed to 'Effective Climate Sensitivity' throughout the manuscript.

*153, Andrews et al. (2012) use years 1 to 150, not 20 to 140 as stated here*

We have deleted the sentence "Usually the slope is computed from year 20 to year 140 of the 4XCO2 experiment (Andrews et al., 2012)"

*171-172, if the method is unbiased, how come nearly all models have negative ZEC at year zero in Figure 2b?*

Figure 2b displayed the 20 year moving average of ZEC. This window crosses over the time emissions cease and hence shows the very bias the method was intended to remove. We have revised Figure 2 to show the unsmoothed ZEC values. Please note that in response to another comment the figure was split into subplots for ESMs and EMICs.

[Figure]

**Figure 2.** (a,c) Atmospheric CO2 concentration anomaly and (b,d) Zero Emissions Commitment following cessation of emissions under the experiment where 1000 PgC was emitted following the 1% experiment (A1). ZEC is the temperature anomaly relative to the estimated temperature at the year of cessation. Top row shows output for ESMs and bottom row for EMICs.

*210, the way efficacy, or epsilon, is calculated it folds in state-dependence of feedbacks into the epsilon. Probably not an essential issue in this paper, since this parameter isn't important for ZEC, though worth a few words of mention*

*210, on the same theme, the study would benefit from a more up-to-date treatment of efficacy. In newer studies this is referred to as time-dependent feedback or pattern effects.*

A paragraph has been added after line 216 to give an extended description of efficacy.

"Efficacy has been shown to arise from spatial patterns in ocean heat uptake (Winton et al., 2010, 2013; Rose et al., 2014), with ocean heat uptake in the high latitudes being more effective at cooling the atmosphere than ocean heat uptake in low latitudes (Rose et al., 2014). This spatial structure in the effectiveness of ocean heat uptake in turn is suspected to originate from shortwave radiation 220 cloud feedbacks (Andrews et al., 2015). The method we have used to calculated efficacy folds state-dependent feedbacks and temporal change in the climate feedback parameter (Rugenstein et al., 2016) into the efficacy parameter."

*218, delete 'of the models'*

This has been done.

*227-229, this sentence is confusing. Models may in addition exhibit cloud adjustments not taken into account in neither Myhre nor Etminan. Therefore models may well lay outside (and they do). I recommend deleting.*

Deleted.

*239-240, somewhat repetitive.*

The sentence did read:

"Some models such as UKESM, CNRM, and UVic exhibit continued warming in the century following cessation of emissions."

There is a typo in the sentence changing its meaning. The sentence should have read:

"Some models such as UKESM, CNRM, and UVic exhibit continued warming in the centuries following cessation of emissions."

This error has been fixed in the revised manuscript.

*241, I believe the authors have mistaken MPI-ESM for Loveclim.*

This is correct. We have adjusted the colour selector to eliminate the second shade of light blue (while maintained a colour-blind friendly palette).

*249-250, any reason we should pay attention to such awkward behaviour?*

The sentence being referred to is:

"The renewed growth in atmospheric $CO_2$ in P. GENIE results from release of carbon from soils overwhelming the residual ocean carbon sink."

This explanation was included for completeness. We were anticipating readers seeing the anomalous behavior of P. Genie and wondering why such a behaviour occurs. In order to not overemphasize this anomaly we have re-written the two sentences into one. The sentence did read:
"One model (P. GENIE) shows a renewed growth in atmospheric $CO_2$ concentration beginning about 600 years after emissions cease. The renewed growth in atmospheric $CO_2$ in P. GENIE results from release of carbon from soils overwhelming the residual ocean carbon sink."

And has been re-written to:

"One model (P. GENIE) shows a renewed growth in atmospheric $CO_2$ concentration beginning about 600 years after emissions cease, resulting from release of carbon from soils overwhelming the residual ocean carbon sink."

*299-344, here I didn't understand why the authors refer to a diagnostic framework (Equation 4), and then don't use it? In particular if ZEC is non-zero, then climate change feedbacks (lambda) will affect the balance, and likewise if there is ocean heat uptake efficacy (non-unitary epsilon).*

The diagnostic framework summarized in Equation 4 was used to compute the components of Figure 7 and Figure 9. It was felt that showing the energy balance terms in graphic form would be easier for readers to interpret than a table of values. To clarify what was done the introduction of the section was re-written from:

"The framework introduced in section \ref{Frame} was applied to the ZECMIP output to partition the energy balance components of ZEC into contributions from the warming effect of the reduction in ocean heat uptake ($-\Delta N$), and the effect of the change in radiative forcing from the ocean and terrestrial carbon fluxes. Figure \ref{EB_bar} shows the results of this analysis for each model averaged over the period 40 to 59 years after emissions cease for the 1000 PgC 1\% (A1) experiment (the same time interval as ZEC$_{50}$)."

To:

"The framework introduced in Section 2.4 was applied to the ZECMIP output to partition the energy balance components of ZEC into contributions from the warming effect of the reduction in ocean heat uptake ($-\Delta N$), and the effect of the change in radiative forcing from the ocean ($F_{ocean}$) and terrestrial carbon fluxes ($F_{land}$). Figure 7 shows the results of this analysis for each model averaged over the period 40 to 59 years after emissions cease for the 1000 PgC 1% (A1) experiment (the same time interval as ZEC50). The components of the bars in Figure 7a are the terms of the right hand side of Equation 4."

*319, delete last instance of 'and'*

Done

*Figure 8, I recommend using a different colour than blue for the outlier, in print it was hard to distinguish.*

The outlier is now shown as a magenta square.

[Figure]

**Figure 9.** Relationship between variables before emissions cease and 50 years after emissions cease. (a) Ocean Heat Uptake (OHU) is computed for 20 year windows with the value at cessation taken from the 1pctCO2 experiment analogous to the how temperature of cessation is computed. (b) Cumulative ocean carbon uptake, (c) Cumulative land carbon uptake. Each marker represents value from a single model. Line of best fit excludes the outlier model IAPRAS which is marked with a magenta square.

*346-355, here I would like to see a statistical test that the slopes are actually nonzero. To me it seems rather random, for instance the slope in panel c would probably disappear if omitting the single high TCRE model.*

We are hesitant here to include a statistical test of slope due to the lack of independence between Earth System Models. ESM have been shown to exhibit an almost phylogenetic relationship with one-another (Knutti et al. 2013). To better explain this we have replaced the sentence:

"However the relationship is weak and several models with high TCRE values have low ZEC values."

With:

"However, these relationships may not be robust due to small number of non-independent models"

*361, use another word than 'Gregory ECS'*

Changed to 'effective climate sensitivity'

*368, any reason to think that ZEC should be related to these quantities? In fact, I think it is great that it is not, since basic theory suggests it shouldn't.*

Upon examining the question in detail it is clear basic theory suggests there should be no relationship. However, before examining the question many of the investigators suspected there might be relationships between ZEC and other metrics. We anticipate that many readers would also intuitively make the same error and hence included the analysis.

*400, the phrasing 'will be required', seems to suggest that sustaining a constant global temperature somehow is desirable. I suggest leaving it to others to decide this.*

We have changed "will" to "would" to make clear that this is a choice.

*418-419, I didn't understand this sentence.*

The sentence has been re-written from:

"However, in experiments with a gradual reduction in emissions the temperature adjustment from the values expected from TCRE (i.e. ZEC) begin to manifest while emissions are ramping-down."

To:

"However, in experiments with a gradual reduction in emissions a mixture of TCRE and ZEC effects occur as the rate of emissions declines."

*425, since we don't expect ZEC to be related to ECS, TCR and TCRE, why start the sentence like this?*

The first part of the sentence "ZEC is poorly correlated to other Earth system metrics, however", has been deleted

*430, I recommend amending: '... cannot be ruled out purely on the basis of models.'*

Done

**Reviewer 2:**

*The study is addressing an important unknown in the climate system as to whether there is continued warming or cooling on a decadal timescale after carbon emissions cease. There is a comprehensive analysis of 18 Earth system models (of either full or intermediate complexity) following a common default experiment of CO2 increasing until 1000 PgC has been emitted and then freely varying.*

*I am very positive towards the ambition, scope and rigour of the manuscript. A key outcome is the message is that ocean and terrestrial carbon uptake is particularly important in determining the Zero Emissions Commitment (ZEC). I have 2 concerns with regards the strength of this conclusion.*

*1. The analysis of the thermal response on a decadal timescale focuses on changes in planetary heat uptake and the efficacy, representing a non-dimensional weighting of planetary heat uptake. The study ignores any explicit discussion of temporal changes in climate feedback parameter (due to their method to diagnose the effective climate sensitivity), particularly associated with clouds. Since the study is focussing on decadal timescales, this omission is likely to be important and should be more fully discussed.*
*Clouds represent one of the biggest uncertainties in climate sensitivity and are known to evolve in time. This potentially important contribution of physical climate feedbacks merits a fuller discussion.*

We have included additional explanation for the efficacy parameter and how we treat it in the analysis, including that any temporal changes in the climate feedback parameter would be folded into our efficacy parameter. After line 216 the following paragraph has been added:

"Efficacy has been shown to arise from spatial patterns in ocean heat uptake (Winton et al., 2010, 2013; Rose et al., 2014), with ocean heat uptake in the high latitudes being more effective at cooling the atmosphere than ocean heat uptake in low latitudes (Rose et al., 2014). This spatial structure in the effectiveness of ocean heat uptake in turn is suspected to originate from shortwave radiation 220 cloud feedbacks (Andrews et al., 2015). The method we have used to

calculated efficacy folds state-dependent feedbacks and temporal change in the climate feedback parameter (Rugenstein et al., 2016) into the efficacy parameter."

*I fully realise that time-variations of the climate feedback parameter may be related to temporal variability in the efficacy. Indeed the authors include a 30% uncertainty in the planetary heat uptake term to take account of this effect, but do not extensively discuss the implications of this uncertainty. In their crucial figure 7, the sign of the ZEC in most model cases is uncertain given the uncertainty arising from the efficacy weighting of the change in the planetary heat uptake. Thus, there needs to be more acknowledgement of the uncertainty in the ZEC due to physical climate feedbacks (or an explanation of why these physical climate feedbacks are less uncertain at the time of zero emission).*
*Whether this uncertainty represents random variability or a systematic trend needs to be addressed.*

Our analysis does show that the time evolution of ZEC is sensitive to physical climate feedbacks. Within our framework, this is captured by the efficacy parameter. To better acknowledge this we have added a paragraph to the discussion about physical climate feedbacks and ZEC. The paragraph has been added to the subsection 'Drivers of ZEC', and reads:

"Our analysis has suggested that the efficacy of ocean heat uptake is crucial for determining the temperature effect from ocean heat uptake following cessation of emissions. Efficacy itself is generated by spatial patterns in ocean heat uptake and shortwave cloud feedback processes (Rose et al., 2014; Andrews et al., 2015). Thus, evaluating how these processes and feedbacks evolve after emissions cease is crucial for better understanding ZEC. As the spatially resolved outputs for ZECMIP are now available (see Section 5), evaluating such feedbacks presents a promising avenue for future research."

*2. There is a larger inter-model spread in the response of the terrestrial carbon feedback. A key conclusion is that the terrestrial carbon response is of central importance in dictating whether surface temperature continues to rise or fall after emissions ceases.*

*However, this conclusion needs to be seriously caveated by the choice of terrestrial carbon cycle and whether nutrient limitation is included. It might be the case that the terrestrial carbon cycle is becoming over strong if there are no constraining limitations applied. It would be useful to group the analysis of the terrestrial response into those model responses with and without nutrient limitation, and then more clearly contrast their behaviour.*

We have conducted an analysis of the terrestrial carbon cycle for models with and without a representation of the nitrogen cycle. The existing section that discussed the nitrogen cycle has been re-written from:

"The remaining models have substantial contributions from both carbon sinks. In all models the reduction in forcing from ocean carbon uptake is smaller than the reduction in ocean heat uptake, suggesting that the post-cessation net land carbon sink is critical to determining ZEC values. Given that the behaviour of the terrestrial carbon cycle varies strongly between models \citep{FriedlingsteinEtAl2006,AroraEtAl2013,AroraEtAl2019} and that many models lack feedbacks such as nutrient limitation and permafrost carbon pools, the strong dependence of ZEC$_{50}$ on terrestrial uptake is concerning. Notably the three ESMs with the weakest modelled terrestrial carbon sink response (ACCESS, MIROC-ES2L, and UKESM) are three which include terrestrial nutrient limitations (Table \ref{MD_ESM_A}, \ref{MD_ESM_B}). The

UVic model includes permafrost carbon and has a relatively weak terrestrial carbon uptake (Table \ref{MD_EMIC_B}). However, Bern and MPI-ESM also have nutrient limitations and have a terrestrial carbon uptake in the middle (Bern) and upper (MPI-ESM) parts of the inter-model range. IAPRAS does not account for either nutrient limitations or permafrost carbon and has the weakest terrestrial carbon uptake of all (Table \ref{MD_EMIC_A}). Ocean carbon uptake also varies substantially between models, with some of the EMICs (P. GENIE, MESM, and IAPRAS) having very high ocean carbon uptake, and two of the ESMs (CanESM5 and CNRM) and having very low ocean carbon uptake."

To:

"The remaining models have substantial contributions from both carbon sinks. In all models the reduction in forcing from ocean carbon uptake is smaller than the reduction in ocean heat uptake, suggesting that the post-cessation net land carbon sink is critical to determining ZEC values. The ocean carbon uptake itself varies substantially between models, with some of the EMICs (P. GENIE, MESM, and IAPRAS) having very high ocean carbon uptake, and two of the ESMs (CanESM5 and CNRM) having very low ocean carbon uptake. Given that the behaviour of the terrestrial carbon cycle varies strongly between models (Friedlingstein et al., 2006; Arora et al., 2013, 2019) and that many models lack feedbacks related to nutrient limitation and permafrost carbon pools, the strong dependence of $ZEC_{50}$ on terrestrial carbon uptake is concerning for the robustness of $ZEC_{50}$ estimates. Notably, the three ESMs, with the weakest terrestrial carbon sink response (ACCESS, MIROC-ES2L, and UKESM), include terrestrial nutrient limitations (Table A1, A2). However, despite including terrestrial nutrient limitation Bern and MPI-ESM simulate a terrestrial carbon uptake in the middle and upper parts of the inter-model range, respectively. The UVic model includes permafrost carbon and has a relatively weak terrestrial carbon uptake (Table A4). IAPRAS does not account for either nutrient limitations or permafrost carbon and has the weakest terrestrial carbon uptake among all models studied here (Table A3)."

A new paragraph has been added to describe our additional analysis of model with and without a terrestrial nitrogen cycle. The paragraph reads:

"To further investigate the effect of nutrient limitation on ZEC we have compared models with and without terrestrial nutrient limitations. Eight of the models that participated in ZECMIP included a representation of the terrestrial nitrogen cycle, ACCESS, CESM2, MIROC-ES2L, MPI-ESM, NorESM, UKESM, Bern and MESM. One model (ACCESS) includes a representation of the terrestrial phosphorous cycle. Figure 8 shows behaviour of the terrestrial carbon cycle before and after emissions cease for models with and without terrestrial nutrient limitations. Figure 8a shows that consistent with Arora et al. (2019) models with a terrestrial nitrogen cycle have on average a lower carbon uptake than those without. However, after emissions cease there is little difference in the terrestrial uptake of carbon between models with and without nutrient limitations. For both sets of model the median uptake is almost the same at 68 PgC and 63 PgC respectively, and the range for models without nutrient limitation fully envelops the range for those with nutrient limitations. Thus, while nutrient limitations do not appear to have a controlling influence on the magnitude of the post cessation terrestrial carbon uptake they have a marked impact on its uncertainty. As with carbon cycle feedbacks (Arora et al., 2019) those models including terrestrial nitrogen limitation exhibit substantially smaller spread than those which do not. This offers hope for future reductions in ZEC uncertainty as more models begin to include nitrogen - and thereafter phosphorus - limitations on the land carbon sink."

The new figure is below:

[Figure]

**Figure 8.** Terrestrial carbon uptake for models with and without a nitrogen cycle, before emissions cease and after emissions cease. Eight models have a representation of nutrient limitations and ten do not. Circles indicate data points for ESMs and triangle indicate data points for EMICs.

*A further recommendation is in the final conclusion is to recap as to how this work compares with prior studies, particularly for the multi-centennial timescale. This context is set out earlier in the motivation, but it is unclear as to the extent of agreement or not with the inferences in the prior work.*

We have added a paragraph after line 427 to discuss how our results compare to prior works. The paragraph reads:

"The results of the ZECMIP experiments are broadly consistent with previous work on ZEC, with a most likely value of ZEC close to zero and a range of possible model behaviours after emissions cease. In our analysis of ZEC we have shown that terrestrial uptake of carbon plays a more important role in determining that value of ZEC on decadal timescales than has been previously suggested. However our analysis is consistent with previous results from Ehlert and Zickfeld (2017) and Williams et al. (2017) in terms of ZEC arising from balance of physical and biogeochemical factors."

*In summary, I think that the study is important and recommend minor edits, particularly to discuss the outcomes of the study in terms of the effect of physical climate feedbacks (via the efficacy) and the controls of nutrient limitation in the terrestrial system, and placing the study in the context of prior work.*

Many thanks for this positive and encouraging review.

***Detailed points:***

*L12 Mention the large uncertainty in the sign of the ZEC from the 30% uncertainty in the efficacy.*

The large uncertainty in efficacy affects our ability to decompose ZEC into energy balance terms for each model, it does not affect the values of ZEC from the models global temperature outputs.

To include the efficacy uncertainty in the abstract we have modified the sentence the describes the analysis from:

"Analysis shows that both ocean carbon uptake and carbon uptake by the terrestrial biosphere are important for counteracting the warming effect from reduction in ocean heat uptake in the decades after emissions cease."

To:

"Analysis shows that both the carbon uptake by the ocean and the terrestrial biosphere are important for counteracting the warming effect from the reduction in ocean heat uptake in the decades after emissions cease. This warming effect is difficult to constrain due to high uncertainty in the efficacy of ocean heat uptake."

*L71/72. Both prior studies are addressing the multi-centennial timescale, rather than the previous discussion of a millennial timescale (L30-41). Your study should relate back to these two prior studies and identify what is different to the arguments outlined by Ehlert and Zickfeld (2017) and Williams et al. (2017).*

To clarify Line 31 has been changed from:
"such that the atmospheric $CO_2$ concentration continues to evolve over several millennia"

To

"such that the atmospheric $CO_2$ concentration continues to evolve over centuries to millennia"

A call-back to Ehlert and Zickfeld (2017) and Williams et al. (2017) has been added to the discussion (see comment above)

"However our analysis is consistent with previous results from Ehlert and Zickfeld (2017) and Williams et al. (2017) in terms of ZEC arising from balance of physical and biogeochemical factors."

*L77 Recommend rephrase to avoid ambiguity so as to make clear to the reader what part of the sentence "only" refers to.*

The sentence did read:
"The ZEC from all emissions over multiple centuries is generally consistent with ZEC from $CO_2$ emissions only for moderate future scenarios (Matthews and Zickfeld, 2012)."

And had been re-written to:

"The ZEC from all emissions over multiple centuries is generally consistent with ZEC from only CO2 emissions, for moderate future scenarios (Matthews and Zickfeld, 2012)."

*P8-11 Recommend placing the model descriptions in Tables 2 to 5 in the Appendix.*

The tables have been moved to appendix A, and are now Tables A1, A2, and A3.

*L175. Add Williams et al. (2017) for the multi-centennial case as that study has addressed the different thermal and carbon controls for delayed warming.*

Done

*P13. In contrast to the model descriptions, I think that the theory in Appendix A could have been placed in the main text, but up to the authors discretion.*

In an effort to make the paper as accessible as possible it was felt that it was better to have the detailed mathematics in an appendix.

*L198 include that the change in ocean heat uptake includes a time-dependent weighting from the efficacy.*

The statement:
"3) the change in ocean heat uptake"

Has been changed to:

"3) the change in effective ocean heat uptake"

*L209. Include that the Gregory ECS is a time average fit over the time period of interest, while the efficacy is time dependent.*

A sentence has been added to line 214:

"Effective climate sensitivity is here calculated as a time average fit and hence is assumed to be a constant, while efficacy values are expected to change in time."

Note 'Gregory ECS' was changed to 'effective climate sensitivity' in response to a comment from Reviewer 1.

*L217 The efficacy values may not simply be representative of internal variability, but may also be associated with systematic shifts in climate feedback, such as systematic changes in cloud types with changing surface temperature.*

We have added a paragraph at this location in the paper to better explain efficacy. See response to general comment.

"Efficacy has been shown to arise from spatial patterns in ocean heat uptake (Winton et al., 2010, 2013; Rose et al., 2014), with ocean heat uptake in the high latitudes being more effective at cooling the atmosphere than ocean heat uptake in low latitudes (Rose et al., 2014). This spatial structure in the effectiveness of ocean heat uptake in turn is suspected to originate from

shortwave radiation 220 cloud feedbacks (Andrews et al., 2015). The method we have used to calculated efficacy folds state-dependent feedbacks and temporal change in the climate feedback parameter (Rugenstein et al., 2016) into the efficacy parameter."

*L223 The authors have acknowledged the importance of the efficacy by including a 30% uncertainty. However, is there a systematic trend to the efficacy or are there random variations in the different models? In diagnostics of ESM2M, there is a progressive increase in the efficacy from close to 1.5 to over 2 in 100 years after emissions cease, which may be equivalently interpreted as a systematic decrease in climate feedback parameter that continues for several centuries; see Williams et al. (2017). If there is a systematic trend, then the implications for the ZEC are different to if there is simply uncertainty in how the efficacy evolves.*

Three of the four EMICs without internal variability all show a declining trend in efficacy, while UVic ESCM shows no change in time (see Figure A1).

We have added a sentence here to note this. The sentence reads:

"Notably efficacy declines in three of the four models, consistent with previous work showing strong trends in efficacy over time (Williams et al., 2017)"

*L254. Clarify the timescale.*

The sentence has been re-written from:

"Of the eight models that extended simulations beyond 150 years, five show temperature peaking then declining (Bern, MESM, DCESS, LOVECLIM and MIROC-ES2L), GFDL shows temperature declining and then increasing but ultimately remaining close to the temperature at cessation, and the UVic model shows continuous, if slow, warming."

To:

"Of the nine models that extended simulations beyond 150 years, seven show temperature on a long-term decline (Bern, MESM, DCESS, IAPRAS, LOVECLIM, P. GENIE, and MIROC-ES2L), GFDL shows temperature declining and then increasing within 200 years after cessation but ultimately remaining close to the temperature at cessation, and the UVic model shows slow, warming."

*Figure 2. It is difficult to pick out individual model types, particularly those coloured blue and green. Recommend split the panels and show the responses for different types of models to gain more insight.*

The figure has been split between ESMs and EMICs. The colour pallet has also been adjusted to remove colours that are too similar.

[Figure]

**Figure 2.** (a,c) Atmospheric CO2 concentration anomaly and (b,d) Zero Emissions Commitment following cessation of emissions under the experiment where 1000 PgC was emitted following the 1% experiment (A1). ZEC is the temperature anomaly relative to the estimated temperature at the year of cessation. Top row shows output for ESMs and bottom row for EMICs.

*L333 The response is interpreted in terms of changes in the "deep ocean circulation".*
*However, the anthropogenic invasion of heat and carbon is being dominated by ventilation of*
*the thermocline, see Sabine et al. (2004) Science or Zanna et al. (2019) PNAS. Schematic*
*figure of Goodwin et al. (2015) Nature Geosciences or model diagnostics in Williams et al.*
*(2017) J. Climate set out this thermocline ventilation view.*

We thank the review for spotting this error and also for pointing us to this additional reference. We agree that the anthropogenic invasion of heat and carbon is mainly dominated by ventilation of the thermocline. We therefore changed the sentence from:

"It has long been suggested that the reason that long-term ZEC was close to zero is compensation between ocean heat and ocean carbon uptake (Matthews and Caldeira, 2008; Solomon et al., 2009; Frölicher and Paynter, 2015), which are both partially controlled by deep ocean circulation (Banks and Gregory, 2006; Xie and Vallis, 2012; Frölicher et al., 2015)."

To:

"It has long been suggested that the reason that long-term ZEC was close to zero is compensation between ocean heat and ocean carbon uptake (Matthews and Caldeira, 2008; Solomon et al., 2009; Frölicher and Paynter, 2015), which are both dominated by the ventilation of the thermocline (Sabine et al., 2004; Banks and Gregory, 2006; Xie and Vallis, 2012; Frölicher et al., 2015; Goodwin et al., 2015; Zanna et al., 2019). "

*L335. The large uncertainty in the efficacy weighting of ocean heat uptake in Figure 7 can change the sign of the ZEC and so this aspect is certainly of comparable importance to the terrestrial carbon sink.*

We have noted this with a few addition sentence added to the end of the paragraph. The sentences read:

"Also notable is the large uncertainty in effective ocean heat uptake, which originates from the uncertainty in efficacy. As efficacy is related to spatial patterns in ocean heat uptake and coupled shortwave cloud feedbacks (Rose et al., 2014; Andrews et al., 2015), shifts in these patterns in time thus likely affect the values of ZEC and hence represents an important avenue for further investigation."

*Figure 10. It is striking how low a proportion of the intermodel variability in the ZEC is explained by any of these metrics. Do these fits improve or alter if different subsets of models are included? The relatively weak fits suggests that the ZEC is being determined by a competition of processes and examining one process in isolation only provides limited insight.*

We did examine fits with ESMs and EMICs alone but the fits were equally bad. As noted by **Reviewer 1** and described in lines 352 to 355 there is no mathematical basis to expect good fits between ZEC and other metrics. However, we have chosen to include the analysis as a priori many readers may suspect such relationships may exist.

*Figure 11 is more encouraging in showing that for the same model type, there is a relationship between the ZEC and the TCRE, with an increasing ZEC with a higher ECS.*

*L368 There is a poor relationship between the ZEC and the TCR and ECS when looking across a range of models, but there is a stronger relationship when looking at the same model.*

The sentence has been re-written from:
"The analysis here has shown that ZEC is poorly correlated to other metrics of climate warming, such as TCR and ECS."

To:

"The analysis here has shown that across models decadal-scale ZEC is poorly correlated to other metrics of climate warming, such as TCR and ECS, though relationships 395 may exist within model frameworks (Figure 12)."

*L399 Mentions that over multiple centuries that warming might further increase or decline. Useful to expand upon that statement and compare further with the prior studies examining the multi-centennial response.*

We have added a paragraph after line 427 to discuss how our results compare to prior works. The paragraph reads:

"The results of the ZECMIP experiments are broadly consistent with previous work on ZEC, with a most likely value of ZEC close to zero and a range of possible model behaviours after emissions cease. In our analysis of ZEC we have shown that terrestrial uptake of carbon plays a more important role in determining that value of ZEC on decadal timescales than has been previously suggested. However our analysis is consistent with previous results from Ehlert and Zickfeld (2017) and Williams et al. (2017) in terms of ZEC arising from balance of physical and biogeochemical factors."

**Addition modification to manuscript:**

In addition to the changes suggested by the reviewers two further modifications have been made to the manuscript.

1) We have now computed TCRE values from the model output provided by each modelling group instead of self-reported values. This change was done due to our reported TCRE values differing from those reported in Arora et al. 2019. This changes Table 7, Figure 10.

2) UVic ESCM results have been updated to results from version 2.9pf to version 2.10 of the model. The results for the two model variants are very similar. All relevant figures, tables and text have been updated.